# TABLEMASTER: A RECIPE TO ADVANCE TABLE UNDERSTANDING WITH LANGUAGE MODELS

**Lang Cao**[I] **Hanbing Liu**[◎]

[I]University of Illinois Urbana-Champaign    [◎]Tsinghua University
langcao2@illinois.edu    liuhb24@mails.tsinghua.edu.cn

## ABSTRACT

Tables serve as a fundamental format for representing structured relational data. While current language models (LMs) excel at many text-based tasks, they still face challenges in table understanding due to the complex characteristics of tabular data, such as their structured nature. In this paper, we aim to enhance LMs for improved table understanding. We identify four key challenges: 1) difficulty in locating target data, 2) deficiency in table semantics, 3) numerical inaccuracies in textual reasoning, and 4) semantic inflexibility in symbolic reasoning. To address these issues, we propose *TableMaster*, a recipe and comprehensive framework that integrates multiple solutions to overcome these obstacles. *TableMaster* first extracts relevant table content and verbalizes it with enriched semantic context. Additionally, we introduce adaptive reasoning, a flexible approach that dynamically adjusts between textual and symbolic reasoning, tailoring the reasoning process to each query. Extensive analyses and experiments demonstrate our findings and the effectiveness of *TableMaster*. On the WikiTQ dataset, *TableMaster* achieves an accuracy of 78.13% using GPT-4o-mini, surpassing existing baselines. We hope this work will serve as a practical step toward more robust and reliable table understanding.

## 1 INTRODUCTION

> *"Data gains extraordinary power as it transcends the simplicity of one dimension to embrace the richness of higher dimensions."*

Tables are widely used in daily life and across various fields, such as healthcare (Ghasemi & Amyot, 2016) and finance (Li et al., 2020; Yi et al., 2025), due to their unique ability to efficiently represent two-dimensional relational data. It is crucial to process tabular data with both efficiency and accuracy. Recently, large language models (LLMs) (Gunasekar et al., 2023; OpenAI, 2024; Touvron et al., 2023) have achieved significant progress in the field of natural language processing. They perform well in a wide range of downstream text-based tasks, including language understanding (Minaee et al., 2024; Zhu et al., 2024) and reasoning (Plaat et al., 2024). Naturally, language models (LMs) are increasingly being used to process and understand tabular data (Fang et al., 2024; Zhang et al., 2024b), enabling reasoning for downstream tasks such as table-based question answering (Pasupat & Liang, 2015) and table-based fact verification (Chen et al., 2020).

However, the data structure of tables inherently possess a unique two-dimensional structure that contrasts with the linear text, which dominates the content in language model pretraining corpora. Most advanced LMs are not specifically optimized for processing tabular data. While techniques such as chain-of-thought prompting (Wei et al., 2023) and other reasoning-enhanced methods (Yao et al., 2023) have enabled LMs to perform satisfactorily in reasoning with linear text, significant room for improvement remains in table-based reasoning (Chen, 2023). A notable gap persists in LMs' ability to fully understand tables and effectively reason with tabular data.

Many previous studies have aimed to improve the table understanding capabilities of LMs. One efficient approach is using prompting to adapt LMs for table understanding without requiring fine-

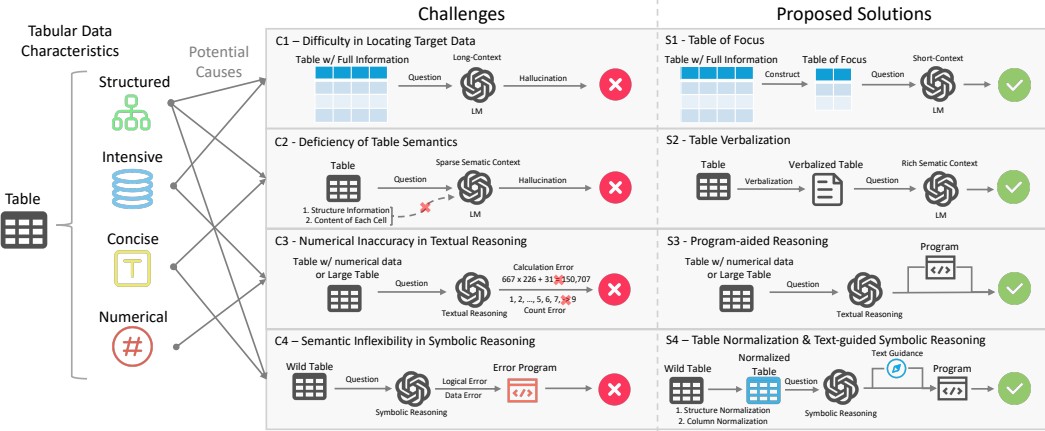

Figure 1: Overview of the challenges and proposed solutions in this work. Tabular data is inherently structured, dense, concise, and numerical. Based on these characteristics, we identify four key challenges. To address them, we propose four targeted solutions. The gray arrows between the characteristics and challenges represent the potential causes of these challenges stemming from specific characteristics. Each proposed solution corresponds to the challenge presented on the left in the same row. *TableMaster* is a unified recipe developed based on these findings.

tuning, making it applicable to any advanced LM. Recent studies primarily adopt two main strategies to enhance table understanding with LMs. The first strategy involves extracting a sub-table that contains relevant content from the original table to reduce the context size, thereby making it easier for LMs to comprehend. Examples include Dater (Ye et al., 2023) and Chain-of-Table (Wang et al., 2024), among others. The second strategy leverages SQL or Python programs to augment numerical reasoning, locate target data, and enhance table understanding of numerical information, as demonstrated by Binder (Cheng et al., 2023) and LEVER (Ni et al., 2023), etc. However, these studies primarily focus on a single basic aspect to enhance the performance of LMs in table understanding or design complex methods with isolated strategies. There is currently an absence of work that provides a systematic and fundamental analysis of table understanding with language models and proposes comprehensive methods for its improvement.

In this paper, we first provide extensive experiments and discussions to identify the challenges in table understanding with language models. To address these challenges, we then introduce *Table-Master*, a recipe and comprehensive framework that integrates multiple solutions to tackle these issues effectively. In summary, this paper makes the following key contributions:

- **Challenges of Table Understanding.** We observe that tabular data is inherently structured, dense, concise, and numerical. Through empirical analysis, we identify four challenges associated with LMs' table understanding: difficulty in locating target data, deficiency of table semantics, numerical inaccuracies in textual reasoning, and semantic inflexibility in symbolic reasoning. (Section 3)

- **A Recipe for Table Understanding.** To address these challenges, we propose targeted solutions: table-of-focus, table verbalization, program-aided reasoning, table normalization, and text-guided symbolic reasoning. Building on these solutions, we introduce a framework as a unified recipe, *TableMaster*. It also incorporates Adaptive Reasoning (AR), a flexible approach that dynamically adjusts between textual and symbolic reasoning, tailoring the reasoning process to each query. (Section 4)

- **Extensive Experiments and Detailed Analyses.** We conduct extensive experiments and provide in-depth analyses to support our findings on table understanding with language models. Furthermore, we evaluate and demonstrate the superior performance of *TableMaster* across three widely used table understanding datasets: WikiTQ, TabFact, and FetaQA. Notably, on the WikiTQ dataset, *TableMaster* achieves an accuracy of 78.13% based on GPT-4o-mini, surpassing existing baselines. (Section 3, Section 5, and Appendix)

## 2 RELATED WORK

**Reasoning with Language Models.** It has been observed that language models (LMs) can exhibit reasoning abilities when they are sufficiently large (Wei et al., 2022; Suzgun et al., 2022). LMs are now widely used for various reasoning tasks, such as question answering (Kamalloo et al., 2023), decision making (Yang et al., 2023), and mathematical reasoning (Ahn et al., 2024). At the inference stage, techniques such as chain-of-thought prompting (Wei et al., 2023) are used to trigger step-by-step reasoning processes and improve reasoning performance. Few-shot prompting (Brown et al., 2020), least-to-most prompting (Zhou et al., 2023), and program-of-thought prompting (Chen et al., 2023) have proven effective in specific scenarios. Methods like self-consistency (Wang et al., 2023b) and structuring the reasoning process in forms like trees (Yao et al., 2023) or graphs (Besta et al., 2024; Cao, 2024a) are also useful for more complex reasoning tasks. Recently, many works have focused on using reinforcement learning (Lightman et al., 2023; Uesato et al., 2022) to improve the reasoning abilities of LMs during training. Our work focuses on inference-time improvements and proposes a general framework applicable to all kinds of LMs for table understanding and reasoning.

**Fine-Tuning LMs for Table Understanding.** Several studies have focused on fine-tuning language models to enhance their understanding of tabular data. For example, based on the masked language modeling approach introduced in BERT (Devlin et al., 2019), models like TaPas (Herzig et al., 2020), Pasta (Gu et al., 2022), and TUTA (Wang et al., 2021) propose specialized pre-training methods to improve LMs' ability to process tables. Similarly, TAPEX (Liu et al., 2022) pre-trains an encoder-decoder model to function as a SQL executor, enabling better table comprehension. Recent advancements, such as TableLlama (Zhang et al., 2024a), TableGPT (Zha et al., 2023), and StructLLM (Zhuang et al., 2024), leverage open-sourced decoder-only models like Llama (Touvron et al., 2023) to pre-train larger models optimized for various downstream table-related tasks. Formula Tuning (Fortune) (Cao et al., 2025) is a reinforcement learning approach that enables language models to perform symbolic table reasoning by deriving executable spreadsheet formulas.

**Adapting LMs for Table Understanding Without Fine-Tuning.** Other studies focus on adapting LMs to table-related tasks without requiring fine-tuning. For instance, Binder (Cheng et al., 2023), LEVER (Ni et al., 2023), and PoTable (Mao et al., 2024) generate SQL or Python programs, extending the capabilities of LMs to analyze tabular data. Dater (Ye et al., 2023), TabSQLify (Nahid & Rafiei, 2024a), ReAcTable (Zhang et al., 2023), TAP4LLM (Sui et al., 2024), and Tree-of-Table (Ji et al., 2024) introduce different methods to construct sub-tables, modifying the tabular context for improved understanding. Chain-of-Table (Wang et al., 2024) generalizes various table operations, dynamically generating reasoning chains to create sub-tables. MIX-SC (Liu et al., 2024b) employs table normalization and leverages self-consistency, combining results from Python agents and textual reasoning to enhance performance. SpreadsheetEncoder (Dong et al., 2024) is specifically designed to interpret tabular data within spreadsheet environments. Our work also follows this direction to focus on adapting LMs without fine-tuning. We identify key challenges in table understanding and address them through our proposed method, which can be applied to any advanced LMs.

## 3 CHALLENGES IN TABLE UNDERSTANDING

As illustrated in Figure 1, we identify and analyze the challenges in table understanding with language models (LMs) through the experiments shown in Figure 2 and related discussions. Additionally, we propose targeted solutions to address these challenges. The detailed settings of the challenge analysis experiment are provided in Appendix D.

**Tabular Characteristics.** Tabular data differs from regular text, which is typically linear and sequential, due to its **structured** nature. Although tabular data can be represented as sequential text, it is fundamentally a two-dimensional array of cells. Each cell primarily contains text, but the cells are interconnected and share relationships with one another. Typically, cells within the same column represent the same feature or type, while cells in the same row correspond to a single data instance. Tables are highly efficient for data representation, often containing a large amount of information, making them inherently data-**intensive**. Moreover, the text in tables is typically **concise**, consisting of simple words and phrases rather than continuous sentences, leading to sparse semantic context.

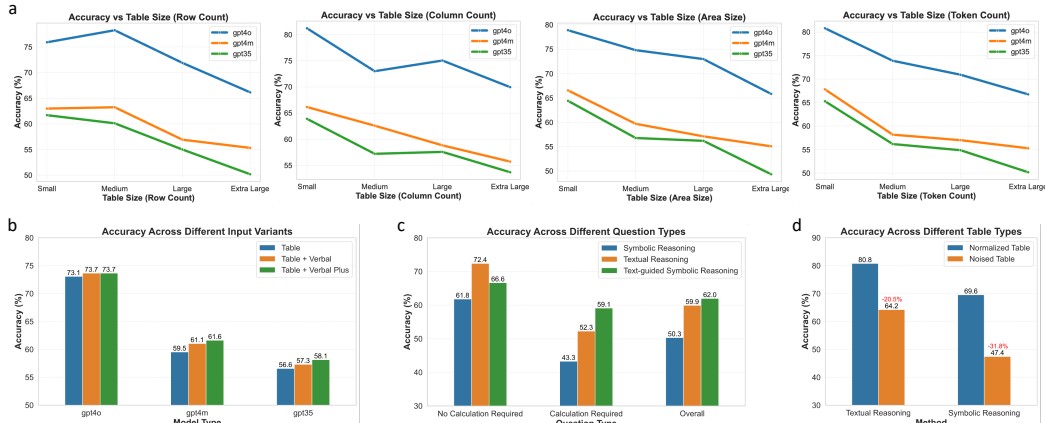

Figure 2: Experimental analysis of challenges in table understanding with language models. (a) Impact of table size on task difficulty. (b) Effect of verbalized tables with enriched semantic context. (c) Performance comparison of different reasoning methods on calculation-required versus non-calculation questions. (d) Performance differences when processing normalized versus noisy tables.

Lastly, tables frequently include substantial amounts of **numerical** data, such as dates, times, scores, and measurements, which often require specialized processing.

## 3.1 DIFFICULTY IN LOCATING TARGET DATA

When LMs encounter tabular data, they often struggle to locate the target data relevant to a given query, leading to misunderstandings. This challenge arises because tabular data is inherently data-intensive, typically containing large volumes of information. Additionally, the structured nature of tabular data makes it challenging for LMs to interpret individual cell contents within the broader context of headers and other structural information. This issue can lead to long-context hallucination (Huang et al., 2024). Moreover, LMs are prone to neglecting information in the middle of the context (Liu et al., 2024a), making it even harder to locate target data and further impairing their overall comprehension of the table. (Figure 1 - C1)

As shown in Figure 2(a), we present the changes in table understanding accuracy across four different table size metrics: row count, column count, area size, and token count, ranging from small to extra-large tables. Row count represents the number of data entries, while column count reflects the number of dimensions or attributes per entry. Area size is the product of row count and column count, and token count refers to table sizes from the perspective of LMs. All figures indicate that, regardless of the model used, overall performance tends to decline as table size increases. For weaker LMs, the performance drop is more pronounced.

To address this, we propose let LMs focusing on specific parts of the table by explicitly constructing a focused sub-table that includes only the relevant information needed for the given context. We define this as the **table-of-focus**. By narrowing the scope, table understanding becomes significantly easier, which aligns with both our previous findings and intuition. (Figure 1 - S1)

## 3.2 TABLE SEMANTIC DEFICIENCY

Tabular data is typically concise, with most cells containing simple words or phrases. Additionally, for each data entry in a row, some descriptive information may reside outside the row, such as in the top header or other structural elements. Understanding a cell in isolation is challenging and often requires a deeper comprehension of the structural relationships within the table. This leads to the problem of **sparse semantic context**, which is fundamentally different from the rich semantic context found in most data used during LMs' pretraining (Dong et al., 2022). The semantic deficiency in tables makes it difficult for LMs to effectively understand and process tabular data. (Figure 1 - C2)

As shown in Figure 2(b), the *Table* represents the case where the LM is provided only with the table input, while the *Table+Verbal* indicates the table along with an additional description, which we refer to as a **verbalized table**. This description is generated by the LMs themselves, whereas *verbal plus* refers to a description produced by more advanced LMs, which can be considered a ground-truth. We observe that verbalization helps LMs perform better on certain tables, leading to a slight overall performance improvement. This effect is more pronounced in weaker LMs, resulting in a 1.5% increase in accuracy. Additionally, the quality of the description plays a crucial role in improvement.

To address this issue, we propose a solution where tables are first verbalized into sequential, natural text as a description and then provided to LMs alongside the original table before they directly tackle table-related tasks. It is similar to table2text (Parikh et al., 2020). This transformation enriches the semantic context, making the data more aligned with the LMs' pretraining, thereby enhancing their ability to effectively understand and process tabular data. (Figure 1 - S2)

### 3.3 NUMERICAL INACCURACY IN TEXTUAL REASONING

Tabular data often contains numerical values, such as dates, times, scores, and other recorded numbers, and is typically intensive. However, when LMs are used to process numerical data in textual reasoning, they often face significant limitations. LMs are prone to arithmetic calculation errors, especially when dealing with large numbers. LMs are also inefficient at handling iterative processes, particularly when the number of iteration steps is large (Chen et al., 2023). (Figure 1 - C3)

As shown in Figure 2(c), questions that do not require calculations are relatively easier, allowing textual reasoning to achieve a strong performance of 72.4%. However, when calculations are required, performance drops significantly, falling below that of the enhanced symbolic reasoning introduced later. Specifically, textual reasoning suffers a 20.1% decline, whereas enhanced symbolic reasoning experiences a more moderate drop of 7.6%.

Symbolic methods offer a promising solution to these challenges and have been explored extensively in prior research (Cheng et al., 2023; Ni et al., 2023; Mao et al., 2024). Using symbolic tools, such as SQL or Python programs in combination with LMs, provides an effective approach to handling numerical data in tabular formats. (Figure 1 - S3)

### 3.4 SEMANTIC INFLEXIBILITY IN SYMBOLIC REASONING

Symbolic methods excel at arithmetic calculations. However, when prompting LMs to generate code for program of thought reasoning, the performance is suboptimal. Instead of truly understanding the context and generating problem-solving code, LMs often rely on memorized code from the pretraining stage (Yang et al., 2024). We refer to this limitation as **semantic inflexibility**. In table understanding, this challenge is exacerbated by the table's complex structure and concise text content. In real-world scenarios, noisy tables further hinder LMs' symbolic reasoning capabilities. Consequently, while symbolic reasoning with numerical data is highly accurate, the generated code may be incorrect due to issues in program logic or data handling, leading to errors or unintended results. (Figure 1 - C4)

As shown in Figure 2(c), basic symbolic reasoning performs worse overall, regardless of whether calculations are required. It indicates that basic symbolic reasoning with current LMs is ineffective. Furthermore, as illustrated in Figure 2(d), when processing the same content in a noisy format, symbolic reasoning suffers a larger performance drop of 31.8%, compared to a 20.5% decline for textual reasoning. This highlights the semantic inflexibility of symbolic reasoning when handling noisy tables.

To address this, we first normalize the table structure and content, ensuring that each column follows a consistent format. We then propose a solution where LMs first engage in textual reasoning before generating symbolic reasoning programs. This preliminary textual reasoning step serves as a guide for subsequent symbolic reasoning, improving alignment with the task context. Our approach can be seen as encouraging LMs to think more thoroughly before reasoning, aligning with techniques like plan-and-solve (Wang et al., 2023a). By incorporating textual reasoning as a foundation, we enhance the accuracy and contextual relevance of symbolic reasoning. As demonstrated in Figure 2(c), this method achieves a higher accuracy of 59.1% for calculation-required questions. (Figure 1 - S4)

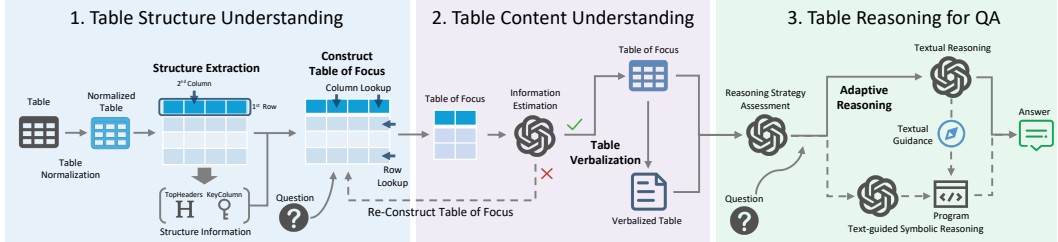

Figure 3: The framework of *TableMaster*. It comprises three stages: (1) table structure understanding, where the table's structure is analyzed, and a table-of-focus is constructed through row and column lookup; (2) table content understanding, where the table-of-focus is reconstructed based on the question, and its information is verbalized to enhance the semantic context; and (3) table reasoning for question answering, where an adaptive reasoning strategy determines whether to use textual reasoning or text-guided symbolic reasoning to derive the final answer. The dashed arrows indicate optional workflows, such as the table-of-focus re-construction and incorporating text-guided symbolic reasoning.

# 4 *TableMaster*: A RECIPE FOR TABLE UNDERSTANDING

Based on findings in Section 3, we introduce a recipe and comprehensive framework, *TableMaster*, as shown in Figure 3. It integrates the propose solution proposed in Section 3 into a unified recipe for table understanding. The framework encompasses three key processes: Table Structure Understanding, Table Content Understanding, and Table Reasoning for QA. All notations are list at Appendix Q.

## 4.1 TASK FORMULATION

In table understanding, the objective is to determine an answer $A$ given a table $\mathbb{T}$ and a question or statement $Q$ related to it. The table $T$ is represented as a two-dimensional array of cells,

$$\mathbb{T}_{m \times n} = \begin{bmatrix} C_{1,1} & C_{1,2} & \cdots \\ C_{2,1} & C_{i,j} & \cdots \\ \vdots & \vdots & \ddots \end{bmatrix}$$

, where $C_{i,j}$ denotes the cell in the $i$-th row and $j$-th column, with the table consisting of $m$ rows and $n$ columns. In table-based question answering tasks, $Q$ represents a question, and $A$ is the expected answer in natural language. In table-based fact verification tasks, $Q$ is a statement about the table's contents, and $A \in \{\text{True}, \text{False}\}$ is a Boolean value indicating whether the statement is correct. Therefore, the goal is to develop a system $\mathcal{F}$ that can predict the answer accurately based on the table and the given question or statement, formalized as $\mathcal{F}(\mathbb{T}, Q) = A$.

## 4.2 TABLE STRUCTURE UNDERSTANDING

The goal of table structure understanding is to analyze the table's structure and construct a Table-of-Focus that contains relevant content for the given question. This process reduces context length and simplifies the table as much as possible.

To enhance the efficiency of the framework, we introduce the **table peek** technique. For structure extraction and certain operations, it is often unnecessary to process the entire table; instead, inspecting only the top rows is sufficient. Given a peek size $k$, the original table $\mathbb{T}_{m \times n}$ is transformed into a peek table $\mathbb{T}_{k \times n}$, where all columns are retained, but the table is truncated to the first $k$ rows.

Given a wild table $\mathbb{T}^W$, we first normalize it. We begin by determining whether the table is in row-major or column-major format. If it is in column-major format, we transpose it using $\mathbb{T} = \text{Transpose}(\mathbb{T}')$. Next, we normalize and clean all columns containing numerical information, ensuring consistency in formats such as dates and numerical values, making them directly processable in bulk by a program. After this normalization process, we obtain the normalized table $\mathbb{T}^N$.

We begin by extracting the top headers $H$ and the key column. The top headers are used for column lookup, while the key column serves as the subject or unique identifier for each row. Next, we prompt LMs to perform column lookup and row lookup to identify the relevant rows and columns required for the task. Specifically, for column lookup, we first define the set of candidate columns as $\mathbb{C} = \text{Rank}(H)$. LMs will also rank all candidates based on their relevance to the question. We then prompt the LMs to select $b$ relevant columns based on a given question $Q$:

$$C^0 = \text{Column Lookup}(\mathbb{T}^N \mid Q),$$

where $C^0 = \{c_i \mid c_i \in H\}$ and $|C^0| = b$. For row lookup, we instruct the LMs to generate an SQL query to efficiently filter and select $a$ relevant rows $R$:

$$R = \text{Row Lookup}(\mathbb{T}^N \mid Q).$$

Using the identified rows and columns, we construct the initial table-of-focus:

$$\mathbb{T}^F_{a \times b} = \text{Table Construction}(\mathbb{T}^N, C^0, R),$$

which contains only the filtered information necessary for the task.

### 4.3 TABLE CONTENT UNDERSTANDING

The goal of table content understanding is to enrich the semantic context of the table.

Studies have shown that LMs can assess whether sufficient information is available to answer a question (Cao, 2024b; Yin et al., 2023). We first prompt the LMs to estimate whether the constructed Table-of-Focus $\mathbb{T}^F_{a \times b}$, containing $C^0$, provides enough information to answer the given question $Q$. If not, additional column attributes from the candidate column set $\mathbf{C}$ are incrementally added from the ranked candidate headers until sufficient information is available or all relevant top headers have been utilized. Subsequently, a total of $a'$ columns from $C$ are selected for further reasoning. We use re-construction to mitigate information loss during the table-of-focus construction process. The detailed re-construction algorithm can be found in Appendix J.

Once the information sufficiency check is passed, we verbalize the table into natural language, adding descriptions to enrich the semantic context and producing a verbalized table:

$$T^{\mathbb{T}} = \text{Verbalization}(\mathbb{T}^F_{a \times b}).$$

This verbalized table is represented as sequential natural language text $T$ essentially rather than a structured table, preserving rich semantic context while maintaining a concise size. This transformation enhances information density, further facilitating the LMs' reasoning for the given question.

### 4.4 TABLE REASONING FOR QUESTION ANSWERING

The goal of this stage is to answer table-related questions by understanding the table precisely and calculating accurately.

We employ an **adaptive reasoning** approach. First, we prompt the LMs to determine the most appropriate reasoning strategy $S$ for the given task. In the instruction, for small tables or those without numerical data, the LMs are allowed to perform textual reasoning directly to derive the final result. For larger tables or those containing numerical data, symbolic reasoning with programmatic execution is selected.

$$S = \text{Strategy Assessment}(\mathbb{T}^F, T^{\mathbb{T}}, Q),$$

where $S \in \{\mathcal{T}, \mathcal{S}\}$ represents the chosen reasoning strategy, with $\mathcal{T}$ denoting textual reasoning and $\mathcal{S}$ denoting symbolic reasoning.

In symbolic reasoning, we first prompt the LMs to perform textual reasoning to generate guidance $G$ without providing the final result. This intermediate reasoning step is then used as input for symbolic reasoning, transitioning to a text-guided symbolic reasoning approach using programmatic methods. This adaptive method dynamically adjusts based on the table's size, complexity, and the nature of the question, ensuring accurate and reliable results.

$$A = \begin{cases} \text{Chain-of-Thought}(\mathbb{T}^F, T^{\mathbb{T}}, Q), & \text{if } S = \mathcal{T} \\ \mathcal{P}(\text{Program-of-Thought}(\mathbb{T}^F, T^{\mathbb{T}}, Q \mid G)), & \text{if } S = \mathcal{S} \end{cases}$$

where chain-of-thought and program-of-thought are two prompting techniques, $\mathcal{P}$ represents a Python or SQL program executor, $A$ is the final answer for the current table understanding task.

Table 1: Performance comparison between *TableMaster* and previous work on WikiTQ and TabFact. The values in the table represent accuracy (%). The best result is **bold**, the second-best result is underlined, and the improvement over the previous best result is highlighted in green. '-' indicates that the result values were not reported in the related papers. For all models in the table, results are obtained from a single inference run without any voting. Our method outperforms all other methods across both datasets and different language models.

| Method | WikiTQ | | | TabFact | | |
|---|---|---|---|---|---|---|
| | gpt-3.5-turbo$_{\sim175B}$ | gpt-4o-mini$_{\sim8B}$ | Llama-3.1$_{70B}$ | gpt-3.5-turbo$_{\sim175B}$ | gpt-4o-mini$_{\sim8B}$ | Llama-3.1$_{70B}$ |
| Text-to-SQL Rajkumar et al. (2022) | 52.90 | - | - | 64.71 | - | - |
| End-to-End QA Wang et al. (2024) | 51.84 | - | - | 70.45 | - | - |
| Few-Shot QA Wang et al. (2024) | 52.56 | - | - | 71.54 | - | - |
| Chain-of-Thought Wang et al. (2024) | 53.48 | - | - | 65.37 | - | - |
| ReAcTable Zhang et al. (2023) | 52.50 | - | - | 74.40 | - | - |
| Binder Cheng et al. (2023) | 56.74 | 58.86 | 50.51 | 79.17 | 84.63 | 78.16 |
| Dater Ye et al. (2023) | 52.81 | 58.33 | 43.53 | 78.01 | 80.98 | 81.57 |
| TabSQLify Nahid & Rafiei (2024a) | 64.70 | 57.02 | 55.78 | 79.50 | 78.75 | 70.70 |
| Chain-of-Table Wang et al. (2024) | 59.94 | 55.60 | 62.22 | 80.20 | 84.24 | 85.62 |
| Tree-of-Table Ji et al. (2024) | 61.11 | - | - | 81.92 | - | - |
| PoTable Mao et al. (2024) | - | 64.73 | 65.56 | - | 88.93 | 87.06 |
| **Ours (*TableMaster*)** | **68.21 (+3.51)** | **78.13 (+13.40)** | **77.95 (+12.39)** | **83.65 (+1.73)** | **90.12 (+1.19)** | **91.16 (+4.10)** |

# 5 EXPERIMENTS

## 5.1 SETTINGS

We conduct extensive experiments to evaluate the performance of *TableMaster*. Specifically, we assess its effectiveness across three different table understanding datasets: WikiTQ (Pasupat & Liang, 2015) (table-based question answering), TabFact (Chen et al., 2020) (table-based fact verification), and FetaQA (Nan et al., 2022) (table-based free-form question answering). For WikiTQ and Tab-Fact, following previous work (Wang et al., 2024; Liu et al., 2024b), we use exact match accuracy as the evaluation metric. For FetaQA, we evaluate performance using BLEU (Papineni et al., 2002) and ROUGE (Lin, 2004) scores. We also conduct experiments on HiTab (Cheng et al., 2022) and FinQA (Chen et al., 2021b). Tables are encoded in Markdown format before being input into language models, with or without addresses, depending on the specific case N.2.

Our experiments utilize OpenAI models hosted on Microsoft Azure. Unless otherwise stated, we set the temperature to 0 to ensure stable output while keeping all other hyperparameters at their default values. The models used in our evaluation include *gpt-4o* (*gpt-4o-0806*), *gpt-4o-mini* (*gpt-4o-mini-0718*), *gpt-3.5-turbo* (*gpt-3.5-turbo-0125*), *o1* (*o1-preview-0912*), and *o1-mini* (*o1-mini-0912*). Additionally, we evaluate our methods on open-sourced *Llama-3.1-70B* (*Llama-3.1-70B-Instruct*). For comparison, we select several strong baselines, including both classic and state-of-the-art methods such as Binder (Cheng et al., 2023), Dater (Ye et al., 2023), and Chain-of-Table (Wang et al., 2024). Performance results for other methods not in this work are cited directly from their original or related papers, with sources indicated alongside the method names in the results table.

Further analysis and additional experiments on *TableMaster* can be found in the Appendix. The prompts used in *TableMaster* can be found in Appendix O, while other prompts used in this work are provided in Appendix P.

## 5.2 MAIN RESULTS

As shown in Table 1, our *TableMaster* approach consistently achieves the highest performance across both WikiTQ and TabFact under different backbone models (*gpt-3.5-turbo*, *gpt-4o-mini*, and *Llama-3.1-70B*). On WikiTQ, *TableMaster* outperforms the strongest baselines by +3.51, +13.40, and +12.39 points, respectively. A similar trend is observed on TabFact, with improvements of +1.73, +1.19, and +4.10 points, demonstrating the robustness of our method across diverse large language models. Results on the FetaQA, HiTab, FinQA dataset are provided in Appendix E.1, E.2, E.3. These results confirm that *TableMaster* not only generalizes well across different base language models but also significantly enhances table understanding and reasoning in complex QA tasks.

Notably, methods such as Binder, Dater, TabSQLify, and Chain-of-Table exhibit subpar performance with *gpt-4o-mini*, in some cases performing worse than with *gpt-3.5-turbo*. Our empirical analysis suggests that these methods primarily rely on symbolic approaches to construct subtables, which often fail to leverage the strengths of chain-of-thought reasoning in textual contexts. This limitation underscores the necessity of integrating advanced textual reasoning strategies, as effectively demonstrated by our *TableMaster* approach.

## 5.3 ABLATION STUDY

To analyze the contribution of each component in *TableMaster*, we conduct an ablation study on WikiTQ and TabFact. Table 2 presents the results, and the performance drop from the full model is highlighted in red. The results demonstrate that removing any component leads to a decrease in accuracy, confirming the importance of each module in the overall framework.

**Structure**. The structure understanding components play a crucial role in table comprehension. Removing structure extraction results in a notable accuracy drop of 3.38% on WikiTQ and 1.14% on TabFact, indicating that explicitly extracting the table's structure is essential for effective reasoning, as failing to do so can lead to errors in subsequent steps. Among lookup strategies, removing row lookup leads to a 1.54% decrease in WikiTQ accuracy, whereas removing column lookup results in a smaller drop of 1.13%. This suggests that row-based information retrieval is more critical than column-based lookup, as large tables typically contain a greater number of rows. Additionally, removing the table-of-focus reduces performance by 1.73% on WikiTQ and 0.79% on TabFact, fur-

Table 2: Ablation results on WikiTQ and TabFact. The values in the table represent accuracy (%), with ▽ indicating the performance drop. The red text highlights the drop magnitude. Removing any component from *TableMaster* results in a decrease in performance.

| Method | WikiTQ | ▽ | TabFact | ▽ |
|---|---|---|---|---|
| *TableMaster* (gpt-4o-mini) | **78.13** | – | **90.12** | – |
| **Structure** | | | | |
| w/o Structure Extraction | 74.75 | (-3.38) | 88.98 | (-1.14) |
| w/o Column Lookup | 77.00 | (-1.13) | 90.51 | (-0.40) |
| w/o Row Lookup | 76.59 | (-1.54) | 89.23 | (-0.89) |
| w/o Table of Focus | 76.40 | (-1.73) | 89.33 | (-0.79) |
| **Content** | | | | |
| w/o Re-Construction | 75.55 | (-2.58) | 89.72 | (-0.40) |
| w/o Verbalization | 75.78 | (-2.35) | 89.23 | (-0.89) |
| **Reasoning** | | | | |
| w/o Textual Reasoning | 73.85 | (-4.28) | 88.39 | (-1.73) |
| w/o Symbolic Reasoning | 76.10 | (-2.03) | 89.18 | (-0.94) |
| w/o Textual Guidance | 75.21 | (-2.92) | 89.67 | (-0.44) |

ther emphasizing its important role in structuring relevant table content to extract key information for reasoning.

**Content**. Table content understanding also significantly influences performance. Eliminating reconstruction, which iteratively refines the Table-of-Focus based on the question, results in a 2.58% accuracy drop on WikiTQ and 0.40% on TabFact, highlighting the importance of this process. Similarly, removing table verbalization, which enriches the semantic context of the table by adding descriptive elements, leads to a 2.35% decrease in WikiTQ accuracy. However, its impact on TabFact is minimal (0.23% drop), suggesting that verbalization becomes even more beneficial for complex table understanding tasks.

**Reasoning**. The reasoning stage exhibits the most significant performance drop when removed. Removing textual reasoning leads to the largest accuracy decline, with a 4.28% drop on WikiTQ and 1.73% on TabFact, underscoring its necessity for complex reasoning tasks. Similarly, removing symbolic reasoning results in a 2.03% and 0.79% drop on WikiTQ and TabFact, respectively, demonstrating that symbolic reasoning enhances numerical and structured table interpretations. Finally, removing textual guidance, which improves the semantic flexibility of symbolic reasoning, reduces accuracy by 2.92% on WikiTQ and 0.44% on TabFact. This highlights that textual guidance is particularly beneficial and important in symbolic reasoning by ensuring alignment with the problem context. More analysis of adaptive reasoning can be found at Appendix L.

## 6 CONCLUSION

In this paper, we explore table understanding with language models. Given the characteristics of tabular data, we identify key challenges in table understanding. To overcome these challenges, we propose *TableMaster*, a recipe and comprehensive framework that integrates multiple solutions. Extensive analyses and experiments demonstrate our findings and the effectiveness of *TableMaster*. In the future, we plan to extend and refine the framework to improve its performance across diverse practical applications, where discussed in Appendix B.

## ACKNOWLEDGMENTS

This work was completed during my internship in the gap year prior to starting my PhD and was motivated by core product needs within the company. I sincerely thank the organization for the resources provided, as well as all the friends and colleagues who supported me during this period.

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

# Contents of Appendix

## A    ETHICS STATEMENT

*TableMaster* introduces a general-purpose, modular framework that improves the ability of language models (LMs) to understand and reason over tabular data. Its applications span domains such as business intelligence, scientific reporting, education, and healthcare, where structured data plays a critical role. By enabling adaptive reasoning across textual and symbolic paradigms, *TableMaster* improves both accuracy and transparency in question answering, verification, and analysis over tables. These advances may lead to better decision-support systems and streamlined human–AI collaboration in spreadsheet-heavy workflows.

Despite these benefits, there are potential risks. As *TableMaster* automates reasoning over tabular content, it could be misused to generate misleading analyses or automate decisions without adequate human oversight. Furthermore, inaccurate reasoning, especially when symbolic operations are applied incorrectly, could result in flawed conclusions or financial misjudgments. Biases in training data might also manifest in generated answers or program logic. In addition, as the framework relies on LLMs' capabilities, disparities across languages, domains, or spreadsheet conventions may lead to uneven performance, potentially disadvantaging users in low-resource settings.

To mitigate these concerns, we propose several safeguards. First, outputs involving symbolic reasoning should be verified via deterministic execution (e.g., code validation or unit tests) before downstream use. Second, we encourage model evaluation on a diverse range of real-world tables, including messy, hierarchical, or multilingual formats. Third, human oversight is recommended in high-stakes applications, especially when deployed in financial, legal, or healthcare settings. Fourth, interpretability tools—such as reasoning traces or program annotations—should be integrated to facilitate debugging and auditing. Finally, we advocate for transparent reporting of model limitations and publishing benchmark results across different domains and table types to promote responsible deployment.

## B    LIMITATIONS, EXTENDABILITY, AND FUTURE WORKS

Although we conduct extensive experiments and in-depth analysis, this work still has certain limitations. However, we believe that *TableMaster* possesses extensibility, allowing for future refinements. These improvements may include technical refinement as well as optimizing its application in downstream applications.

### B.1    TECHNICAL REFINEMENT

**Wild Table.** In our experiments, the tables in the three datasets we use are already cleaned; therefore, we do not explicitly implement table normalization in our evaluation experiments. However, we conduct analysis experiments to highlight the importance of table normalization for handling wild tables. In practical scenarios, various tools are available for table normalization. Regular expression matching can be employed for formatting, and small language models can also be leveraged to efficiently process and normalize tables (Nahid & Rafiei, 2024b).

**Hierarchy Table.** In our work, we assume all tables are flat, allowing for straightforward utilization and extraction of structural information. However, many real-world tables are hierarchical, where data is organized in a tree structure, making table structure understanding more challenging. We envision two possible solutions: converting hierarchical tables into flat tables or designing a tree-based structure extraction method to effectively locate target data.

**Table-of-Focus Construction.** In designing the Table-of-Focus, we employ two efficient methods: LM prompting for column lookup and SQL generation for row lookup. The Table-of-Focus is then constructed based on the results of these two lookups. Many previous works (Ji et al., 2024; Wang et al., 2024) have introduced complex approaches for extracting relevant sub-tables. In contrast, our method remains intentionally simple, prioritizing efficiency and adaptability. We believe that in the future, more advanced techniques may emerge to further enhance the extraction of key information.

**Table Verbalization.** To facilitate the implementation of *TableMaster*, we utilize language models themselves to verbalize the table. However, the quality of the generated text is not optimal due to the challenges of open-ended text generation. Several existing studies, such as Table-to-Text (Parikh

et al., 2020), have explored this sub-task. In the future, we can enhance performance and efficiency by replacing this step with specifically trained small language models, which could further improve the semantic density of the verbalized table.

**Adaptive Reasoning.** Adaptive reasoning can be unstable, as language models may not always select the optimal strategy. We further explore this issue in Appendix L. In the future, training a dedicated machine learning model to guide LMs in selecting the most effective reasoning strategy could improve stability and performance.

**Information Missing.** The construction of the Table-of-Focus involves a trade-off between precision and recall. If recall is insufficient, essential information may be missing for final reasoning, while low precision can render the extracted content less useful. Although we use re-construction to mitigate information loss during the Table-of-Focus construction process, our analysis reveals that some information missing persist in row lookup. We further investigate this issue in Appendix M.

**Efficiency.** Efficiency is crucial in table processing and table understanding. To enhance efficiency, we incorporate the table peek technique, which reduces the context that language models need to process at certain steps. We further explore this technique in Appendix H and analyze the overall efficiency in Appendix I. In real-world applications, for optimal efficiency, we consider replacing certain steps with specialized small language models, balancing the trade-off between efficiency and performance .

### B.2 Downstream Applications

**Web Tables.** The web contains a vast number of structured tables, including Wikipedia tables, government reports, and other online tabular data. Extracting and reasoning over these tables is crucial for applications such as fact verification, web search, and knowledge graph construction. *TableMaster* enhances the ability to interpret, query, and reason over these tables, enabling more accurate and context-aware information retrieval.

**Spreadsheets.** Spreadsheets are widely used in business, finance, and scientific research for data management and analysis. Traditional spreadsheet tools require manual formula creation and human intervention to derive insights. In contrast, *TableMaster* can automate tasks such as data summarization, trend analysis, anomaly detection, and reasoning-based computations. By integrating with tools like Microsoft Excel and Google Sheets, *TableMaster* enables intelligent spreadsheet interactions, allowing users to query data using natural language and receive precise, structured responses.

**Databases.** Structured databases store vast amounts of relational data, typically accessed through SQL queries or predefined interfaces. However, many users lack SQL proficiency, posing barriers to efficient data retrieval. *TableMaster*, with its Table-of-Focus mechanism, facilitates the quick understanding of large databases, enabling seamless querying of relational data without the need for manual SQL query writing. Additionally, it enhances database reasoning tasks, including knowledge extraction, making structured data more accessible to non-technical users.

In real-world applications, different scenarios have varying requirements, and it may not be necessary to incorporate all aspects of *TableMaster*. Instead, certain components can be adapted or selectively applied based on specific needs.

Finally, as discussed above, there is still much work to be done in the future to further enhance language model-based table understanding. We hope this work serves as a recipe of comprehensive references on current state-of-the-art methods and provides guidance for future advancements in this field.

## C DATASETS USED FOR EVALUATION

Table 3 shows all datasets use for evluation in this study, license and source are also included.

## D DETAILED SETTINGS OF CHALLENGE ANALYSIS EXPERIMENTS

We conduct extensive experiments to analyze the challenges of table understanding with language models (LMs). Specifically, we perform challenge analysis experiments on the WikiTQ dataset

Table 3: Benchmarks used for evaluation.

| Dataset | # Test | Table Type | Domain | License | Source |
|---|---|---|---|---|---|
| WikiTQ (Pasupat & Liang, 2015) | 4,217 | Relational | Wikipedia | CC-BY-SA-4.0 | Link |
| TabFact (Chen et al., 2020) | 2,024 | Relational | Wikipedia | CC-BY-4.0 | Link |
| FetaQA (Nan et al., 2022) | 1,165 | Relational | Wikipedia | CC-BY-SA-4.0 | Link |
| FinQA (Chen et al., 2021b) | 1,147 | Relational | Finance | MIT | Link |
| HiTab (Cheng et al., 2022) | 1,583 | Hierarchical | Reports | C-UDA 1.0 | Link |

(Pasupat & Liang, 2015), which consists of 4,344 data instances. Following previous work (Wang et al., 2024; Liu et al., 2024b), we use the exact match of the final answer as the evaluation metric to measure accuracy. Our experiments utilize OpenAI models hosted on Microsoft Azure[1]. Unless otherwise stated, we set the temperature to 0 to ensure stable output while keeping all other hyperparameters at their default values. For each model, we use the following versions: *gpt-4o* (*gpt-4o-0806*), *gpt-4o-mini* (*gpt-4o-mini-0718*), *gpt-3.5-turbo* (*gpt-3.5-turbo-0125*), and *o1* (*o1-preview-0912*).

**Effect of Table Size** (Figure 2(a)). We evaluate how table size impacts task difficulty using a direct prompting approach (Prompt 21) with *gpt-4o*, *gpt-4o-mini* and *gpt-3.5-turbo* to generate answers. We categorize table size based on four metrics: row count, column count, area size (computed as the product of row and column counts), and token count (measured using the *cl100k_base* encoding). The tables are divided into four size categories—small, medium, large, and extra-large—strictly partitioned into quartiles from the smallest to the largest. We then analyze results by splitting performance based on table size.

**Effect of Verbalization** (Figure 2(b)). We investigate the impact of enriching semantic context through verbalized tables by comparing three approaches. In the *Table* setting, the LM processes the raw table directly using direct prompting (Prompt 21). In *Table + Verbal*, the table is first verbalized using the LM itself (Prompt 24), and both the original and verbalized tables are then provided as input. Lastly, in *Table + Verbal Plus*, the verbalized table is generated using *gpt-4o*, further enhancing the semantic richness of the input.

**Comparison of Reasoning Methods** (Figure 2(c)). We compare different reasoning approaches—textual reasoning (Prompt 22), symbolic reasoning (Prompt 23), and text-guided symbolic reasoning (Prompt 25)—on calculation-required versus non-calculation questions using *gpt-3.5-turbo*. To classify WikiTQ questions into calculation-required or not, we use *o1* (Prompt 28), identifying 2,692 calculation-required questions and 1,652 non-calculation questions. The results are then analyzed based on this classification.

**Impact of Noisy Tables** (Figure 2(d)). We investigate how performance varies between normalized and noisy tables. To generate noisy tables, we use *o1* (Prompt 29), instructing it to introduce noise into table contents while preserving actual values and diversifying entries within columns. Additionally, each table has a 50% chance of being randomly transposed from the default row-major format to the column-major format. We then filter the generated tables through a combination of human verification and *o1* checks to ensure that answers remain derivable from the noisy tables. After filtering, 2,565 noisy tables remain. We evaluate textual reasoning (Prompt 22) and symbolic reasoning (Prompt 23) on both the noisy and original normalized tables using *gpt-4o-mini*.

# E  EXTENDED EXPERIMENTS ON ADDITIONAL TABLE UNDERSTANDING BENCHMARKS

We perform additional experiments on diverse table-understanding tasks to further assess the robustness of *TableMaster*.

## E.1  EVALUATION ON FREE-FORM QA WITH THE FETAQA DATASET

PaLM 2 has been deprecated (Anil et al., 2023) and is no longer accessible. Therefore, we use a comparable language model, *gpt-4o*, to conduct experiments on FetaQA and compare the results

---

[1]https://azure.microsoft.com/en-us/support/legal/

Table 4: Performance comparison on FetaQA. The values are multiplied by 100, and the percentage improvement represents the performance gain compared to the end-to-end QA of the base model. The results demonstrate that *TableMaster* achieves strong performance in long-form question answering.

| Methods | BLEU | ROUGE-1 | ROUGE-2 | ROUGE-L |
|---|---|---|---|---|
| Fine-Tuning (T5-large) (Ye et al., 2023) | 30.54 | 63 | 41 | 53 |
| End-to-End QA (Codex) (Chen et al., 2021a) | 27.96 | 62 | 40 | 52 |
| End-to-End QA (PaLM 2) (Wang et al., 2024) | 28.37 | 63 | 41 | 53 |
| Dater (PaLM 2) (Ye et al., 2023) | 29.47 | 63 | 41 | 53 |
| Chain-of-Table (PaLM 2) (Wang et al., 2024) | 32.61 (+14.9%) | 66 (+4.8%) | 44 (+7.3%) | 56 (+5.7%) |
| End-to-End QA (gpt-4o) | 24.91 | 62.05 | 41.29 | 50.36 |
| **Ours (Tablemaster - gpt4o)** | **28.94 (+16.2%)** | **66.06 (+6.5%)** | **45.29 (+9.7%)** | **54.56 (+8.3%)** |

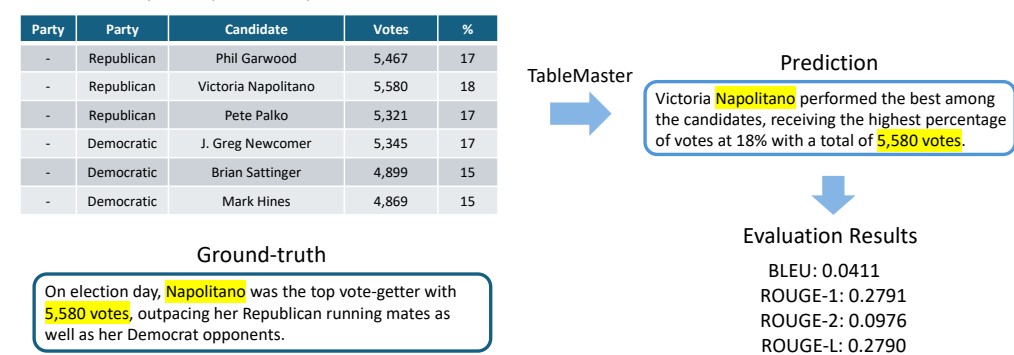

Figure 4: An example (fetaqa-164) from the FetaQA dataset where the result is accurate, but the evaluation metric assigns a low score.

with previous methods. Additionally, we use 20 exemplars for few-shot in-context learning to align with the dataset's format.

Table 4 shows that *TableMaster* improves free-form question answering performance on FetaQA compared to the base End-to-End QA model, achieving improvements of 16.2% in BLEU and 6.5% in ROUGE-1. These improvements surpass those of Chain-of-Table when compared to its respective End-to-End QA baseline.

However, the improvement of *TableMaster* over baseline methods remains marginal, with some values even falling below those of previous approaches in absolute terms. We believe this does not fully reflect the model's actual performance in free-form QA. We attribute this to the n-gram text similarity metrics used in ROUGE-1/2/L (Lin, 2004), which are known to be insensitive to improvements gained from in-context learning (Maynez et al., 2023). These metrics struggle to capture stylistic and structural enhancements in free-form text generation. Since models rely on instructions and a limited number of examples, they may not fully adapt to the expected output format, leading to an underestimation of performance gains.

To further investigate this, we analyze a specific case, FetaQA-164, as shown in Figure 4. In this instance, the BLEU and ROUGE metrics assign low scores, as only two words match in the entire sentence. However, manual review confirms that the generated answer is indeed correct—these two words are the most important, and the overall meaning of the response is both accurate and superior to the ground truth. This highlights the limitations of ROUGE in evaluating free-form QA and suggests that qualitative analysis is essential for a more comprehensive assessment of model improvements. Nonetheless, based on quantitative analysis, *TableMaster* is overall effective.

### E.2 EVALUATION ON HIERARCHICAL TABLES WITH THE HITAB DATASET

UUnlike WikiTQ and TabFact, the HiTab dataset (Cheng et al., 2022) contains *hierarchical* tables that violate the flat row–column assumption. Such structure challenges models to reason across multi-level headers and parent–child cell relations.

We employ a *GPT-4o* backbone and follow HiTab's official data split. MultiCoT (CYQIQ, 2025) extends **Chain-of-Table** reasoning to multiple tables; both MultiCoT and *TableMaster* operate on the same converted relational tables. E5 (Zhang et al., 2024c) represents the current SOTA on HiTab, being explicitly tailored for complex tables. Because our framework (*TableMaster*) is designed for relational tables, we introduce a lightweight *relational-table converter* via prompting *o1*: each hierarchical table is decomposed into several relational subtables, while contextual tags are propagated to preserve structural cues.

Table 5: Accuracy (%) on the HiTab dataset. "After Converting" rows evaluate models on relational tables produced by our converter; "Direct" reports results on the original hierarchical tables.

| Method | Accuracy |
|---|---|
| *After Converting to Relational Tables* | |
|     MultiCoT (original (CYQIQ, 2025)) | 64.0 |
|     MultiCoT (optimized prompt) | 70.0 |
|     MultiCoT (optimized prompt + verbalized table) | 73.5 |
|     ***TableMaster*** | **74.2** |
| *Direct* | |
|     E5 (Zhang et al., 2024c) | 77.3 |

Table 5 shows that *TableMaster* outperforms the strongest chain-of-table baseline by +0.7 point (from 73.5 to 74.2). Residual errors are mainly due to information loss during the conversion step. Future work will integrate hierarchical relations directly into the reasoning module.

### E.3 EVALUATION ON NUMERICAL REASONING WITH THE FINQA DATASET

FinQA (Chen et al., 2021b) requires multi-step numerical reasoning over financial reports—e.g., computing growth rates or combining multiple cells with arithmetic operators. Hence it evaluates whether *TableMaster* can execute numerical formulas correctly, not just extract text spans. We keep the same training recipe as in. Two backbones are considered: *GPT-4o-mini* (4m) and *GPT-4o* (4o).

Table 6: Accuracy on FinQA. *TableMaster* consistently boosts numerical-reasoning accuracy over both backbones.

| Method | Accuracy (%) | Δ |
|---|---|---|
| GPT-4o-mini | 50.7 | – |
| ***TableMaster*** (4m) | **66.4** | +15.7 |
| GPT-4o | 63.1 | – |
| ***TableMaster*** (4o) | **70.9** | +7.8 |

Table 6 shows that *TableMaster* delivers impressive improvements on both backbones, demonstrating that its symbolic reasoning module reliably handles complex calculations in the financial domain.

## F TABLE UNDERSTANDING BASELINES

To better facilitate future research, we evaluate different reasoning methods across various base models. Table 7 presents the accuracy results of our reproduced baselines on WikiTQ and TabFact, comparing different base LLMs and reasoning methods. The table includes evaluations on *o1-preview*

Table 7: Results of our reproduced baselines on WikiTQ and TabFact. The values in the table represent accuracy (%).

| Base LLM | Method | WikiTQ | TabFact |
|---|---|---|---|
| o1-preview$_{\sim300B}$ | Direct | 84.60 | 92.05 |
| o1-mini$_{\sim100B}$ | Direct | 83.49 | 91.35 |
| gpt-4o$_{\sim200B}$ | Direct | 73.07 | 84.73 |
| | Chain of Thought | 83.98 | 91.90 |
| | Program of Thought | 74.63 | 90.02 |
| | *TableMaster* (gpt-4o) | **84.55** | **94.52** |
| gpt-4o-mini$_{\sim8B}$ | Direct | 59.53 | 71.25 |
| | Chain of Thought | 72.97 | 87.40 |
| | Program of Thought | 61.83 | 85.18 |
| | *TableMaster* (gpt-4o-mini) | **78.13** | **90.12** |
| gpt-3.5-turbo$_{\sim175B}$ | Direct | 56.58 | 70.90 |
| | Chain of Thought | 59.92 | 69.52 |
| | Program of Thought | 50.32 | 68.82 |
| | *TableMaster* (gpt-3.5-turbo) | **68.21** | **83.65** |

($\sim$300B), *o1-mini* ($\sim$100B), *gpt-4o* ($\sim$200B), *gpt-4o-mini* ($\sim$8B), and *gpt-3.5-turbo* ($\sim$175B). The exact number of parameters for several LMs (e.g., GPT, o1) has not been publicly disclosed. Most parameter counts are estimates reported to provide context for understanding model performance. For more precise information, please refer to the original or future official documentation (Abacha et al., 2025). Each model is tested with various reasoning strategies, including Direct, chain of thought, and Program of Thought, alongside our proposed *TableMaster*.

Across all base models, TableMaster consistently achieves the highest accuracy. For gpt-4o, TableMaster reaches 84.55% on WikiTQ and 94.52% on TabFact, outperforming both chain of thought (83.98%, 91.90%) and Program of Thought (74.63%, 90.02%). Similarly, for gpt-4o-mini, TableMaster achieves 78.13% on WikiTQ and 90.12% on TabFact, significantly improving over the Direct method (59.53%, 71.25%) and surpassing chain of thought (72.97%, 87.40%).

The performance gap is even more pronounced for gpt-3.5-turbo, where TableMaster reaches 68.21% on WikiTQ and 83.65% on TabFact, significantly outperforming both chain of thought (59.92%, 69.52%) and Program of Thought (50.32%, 68.82%). Interestingly, we observe that while *TableMaster*'s improvement is limited on gpt-4o, the weaker the base model, the greater the performance improvement. While o1-preview and o1-mini achieve high accuracy with the Direct method (84.60%, 92.05% for o1-preview and 83.49%, 91.35% for o1-mini), the results of *TableMaster* on gpt-4o demonstrate that our method is capable of achieving state-of-the-art performance across different LL architectures.

Additionally, we find that chain of thought reasoning is highly effective, achieving strong accuracy across models. Even a simple chain of thought approach outperforms previous methods that rely solely on symbolic reasoning (Mao et al., 2024), indicating that chain of thought should be retained as a key component in the reasoning framework.

These results confirm that *TableMaster* enhances table reasoning performance across various LLMs, effectively outperforming both direct prompting and traditional reasoning strategies, particularly in cases where table complexity and reasoning demands are higher.

## G  PERFORMANCE ANALYSIS UNDER DIFFERENT TABLE SIZES

Table 8 presents a performance comparison across different table sizes, categorized into small (<2k tokens), medium (2k$\sim$4k tokens), and large (>4k tokens). The results compare several methods, including Binder (Cheng et al., 2023), Dater (Ye et al., 2023), and Chain-of-Table (Wang et al.,

Table 8: Performance Comparison Across Table Sizes (Token).

| Method | Table Size (Token) | | |
| --- | --- | --- | --- |
| | Small (<2k) | Medium (2k ∼ 4k) | Large (>4k) |
| Binder (Cheng et al., 2023) | 56.54 | 26.13 | 6.41 |
| Dater (Ye et al., 2023) | 62.50 | 42.34 | 34.62 |
| Chain-of-Table (Wang et al., 2024) | 68.13 | 52.25 | 44.87 |
| *TableMaster* (gpt-3.5-turbo) | **69.01** | **58.00** | **56.73** |
| *TableMaster* (gpt-4o-mini) | 78.71 | 70.50 | 70.19 |

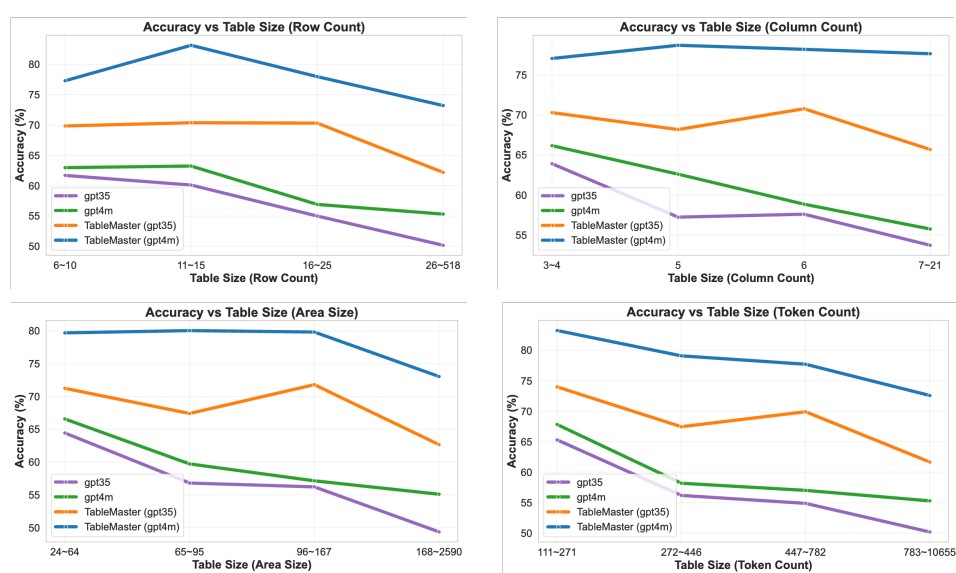

Figure 5: Performance Comparison Across Table Sizes (Row Count, Column Count, Area Size, Token Count).

2024), against *TableMaster*. All methods are evaluated using *gpt-3.5-turbo*, with additional results of *TableMaster* provided for *gpt-4o-mini*.

Across all table sizes, *TableMaster* consistently outperforms baseline methods. Specifically, for *gpt-3.5-turbo*, *TableMaster* achieves the highest performance in all table size categories, scoring 69.01% on small tables, 58.00% on medium tables, and 56.73% on large tables. This demonstrates its ability to maintain robust performance even as table size increases, significantly outperforming Binder, Dater, and Chain-of-Table, especially on medium and large tables, where the performance gap becomes more pronounced.

Furthermore, *TableMaster* with *gpt-4o-mini* achieves even stronger performance, with accuracy scores of 78.71% (small tables), 70.50% (medium tables), and 70.19% (large tables). These results highlight that leveraging stronger base models further enhances *TableMaster*'s effectiveness, making it particularly well-suited for large-scale table reasoning tasks. Notably, when transitioning from medium to large tables, *TableMaster* (*gpt-4o-mini*) experiences only a 0.31% performance drop (from 70.50% to 70.19%), demonstrating its strong capability in handling increasing table complexity. This minimal decline contrasts sharply with other methods, which show significantly larger drops, further reinforcing the scalability and robustness of *TableMaster* in processing large-scale tabular data.

Figure 5 illustrates the accuracy trends of different models across various table sizes, categorized based on row count, column count, area size, and token count. The models evaluated in this study include *gpt-3.5-turbo* (*gpt35*), *gpt-4o-mini* (*gpt4m*), *TableMaster* (*gpt35*), and *TableMaster* (*gpt4m*). The results provide insights into how these models handle increasing table complexity and size,

revealing the comparative strengths and limitations of each approach. The size split in this study is strictly partitioned into quartiles, ranging from the smallest to the largest tables.

**Row Count.** The top-left plot analyzes accuracy trends as row count increases. *TableMaster* (*gpt4m*) consistently outperforms other models, maintaining high accuracy levels even with an increasing number of rows. In contrast, *gpt-3.5-turbo* (*gpt35*) starts with the highest accuracy, peaking in the 11–15 row range before experiencing a decline as row count further increases. Smaller models such as *gpt35* and *gpt4m* exhibit a sharper decline, highlighting the challenge of processing larger tables with more rows.

**Column Count.** The top-right plot examines model performance as column count increases. *TableMaster* (*gpt4m*) again achieves strong performance, peaking at around five columns before showing a slight decline. This result highlights the effectiveness of **table-of-focus re-construction**, demonstrating that column re-selection can effectively adapt to scenarios with many columns. While *gpt35* initially maintains the highest accuracy, other models experience a steeper drop as the number of columns increases. These trends suggest that column-heavy tables pose greater challenges for reasoning compared to row-heavy tables, likely due to the increased dimensional complexity and interdependencies between attributes.

**Area Size.** The bottom-left plot evaluates the relationship between accuracy and table area size, calculated as the product of row and column counts. *TableMaster* (*gpt4m*) reaches peak performance in the mid-range (96–167 area size) before slightly declining for larger tables. *gpt35* initially performs well but deteriorates as table area size increases, while *gpt4m* and *gpt35* show a noticeable decline overall, reinforcing that larger tables significantly impact accuracy across models.

**Token Count.** The bottom-right plot assesses accuracy as a function of table token count, which reflects the amount of textual information models need to process. *TableMaster* (*gpt4m*) consistently achieves the highest accuracy, followed by *TableMaster* (*gpt35*). A general downward trend is observed across all models as token count increases, indicating that larger input lengths negatively affect performance. Notably, *gpt35* experiences the sharpest drop, suggesting its lower capacity for handling long-context table data compared to *gpt4m*.

Overall, these findings confirm that *TableMaster* is highly scalable and generalizable across different table sizes, consistently outperforming previous methods, particularly in handling larger and more complex tables. Its robust performance and gradual decline in accuracy as table size increases make it a reliable and efficient solution for table-based reasoning tasks.

## H  PERFORMANCE ANALYSIS UNDER DIFFERENT TABLE PEEK SIZES

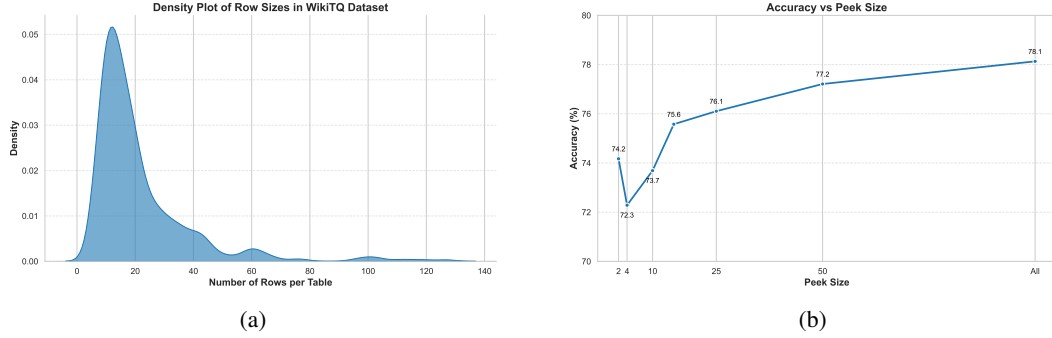

(a)                                                                 (b)

Figure 6: The row count distribution in the WikiTQ dataset and the analysis of accuracy variation with different peek sizes.

We propose the concept of table peek, which enhances the efficiency of *TableMaster* for table understanding tasks by reducing the context that language models need to process at certain steps.

To analyze the effectiveness of this approach, we first examine the row count distribution in the WikiTQ dataset, as shown in Figure 6a. To improve visualization, we remove 72 extreme outliers with exceptionally large row counts. The resulting density plot illustrates that the majority of tables con-

tain fewer than 20 rows, with a pronounced peak around 10 rows. As the number of rows increases, the density gradually declines, indicating that large tables are relatively uncommon. Although a small number of tables exceed 100 rows, their frequency is minimal.

The line graph in Figure 6b illustrates the variation in accuracy with different peek sizes, where the peek size determines the number of rows considered during processing. Initially, accuracy is relatively low when only a small number of rows (e.g., 2–4) are used, reaching its minimum at a peek size of 4. We hypothesize that this occurs because, at a peek size of 2, the table includes only the top headers and a single example row, which may provide a clear structure for the language model to follow. However, at a peek size of 4, the table includes three example rows, potentially causing the language model to overfit the first few rows and misinterpret the overall table structure. This misalignment may lead to ineffective SQL generation for row lookup, resulting in a temporary drop in accuracy.

As the peek size increases, accuracy improves significantly, showing a sharp rise up to 25 rows. Beyond this point, the accuracy continues to improve but at a slower rate, eventually reaching its peak when the entire table is utilized ('All'). This trend suggests that a moderate peek size can effectively balance efficiency and accuracy, eliminating the need to process the full table while still maintaining strong performance.

# I    EFFICIENCY ANALYSIS OF *TableMaster*

## I.1    THEORETICAL ANALYSIS

Efficiency is a critical factor in table-understanding methods. We analyze the efficiency of *TableMaster* theoretically, following the notations introduced in Section 4. Our analysis considers the length of the table input as the primary computational cost, excluding any additional prompts or external information, and does not account for output length. This is because, in most cases, the output is relatively short compared to the large volume of data in the table. Specifically, we define the computational cost in terms of the total area size of the table that the language model processes.

Below are the main components of our efficiency analysis:

- **Structure extraction:** $k \times n$
- **Row lookup:** $k \times n$
- **Column lookup:** $n$
- **Table-of-Focus Re-Construction** $a \times b \times e$
- **Table Verbalization:** $a \times b$,
- **Reasoning Strategy Assessment:** $a \times b$,
- **Reasoning:** $1.5\,a \times b$ (where the factor 1.5 accounts for textual processes weighted as 1 and symbolic processes weighted as 2)

Here, $k$ represents the size of table peek, and $e$ represents the number of table-of-focus reconstructions after information estimation. $a$ and $b$ denote the dimensions of the table-of-focus $\mathbb{T}_{a \times b}$. Combining these components, the total computational cost is given by:

$$\text{Total Cost} = (2\,k + 1) \times n + (e + 2.5) \times (a \times b). \tag{1}$$

## I.2    EMPIRICAL ANALYSIS

The bar chart in Figure 7 illustrates the change in table area size before and after table condensation for the WikiTQ and TabFact datasets. The y-axis represents the table size, while the x-axis categorizes the datasets. Each dataset has two bars: the blue bar represents the original table size, and the orange bar represents the condensed table size after table-of-focus construction. WikiTQ exhibits a significant reduction in table size, approximately 1:3, with the condensed table being much smaller than the original. In contrast, TabFact also undergoes condensation but to a lesser extent, around

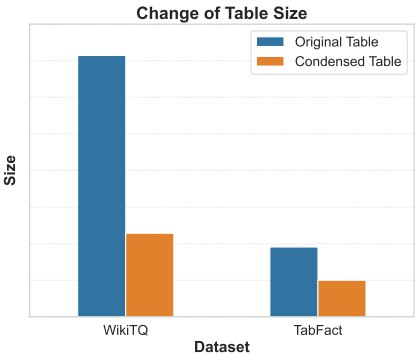

Figure 7: Changes in Table Condensation After Table-of-Focus Construction in Table Structure Understanding.

1:2. This suggests that WikiTQ tables require more substantial structural modifications to focus on relevant content, while TabFact tables need comparatively less condensation.

As shown in Equation 1, the theoretical cost is independent of the number of rows $m$, while $a \times b$ reflects the size of the small sub-table, which is influenced by the estimated table condensation ratio 2.5. As stated in Table 10, the reconstruction occurs 1.5 averagely, so $e$ is typically 1.5. In an ideal scenario, if the peek size is negligible, the cost is approximately $1.6 \times (m \times n)$. In the worst-case scenario, where the entire table must be examined and all content is required, the cost reaches $6 \times (m \times n)$ approximately. The estimation range for each table is 1.6 to 6 times the original table size.

Recent advancements in table understanding, such as CHAIN-OF-TABLE (Wang et al., 2024) and TREE-OF-TABLE (Ji et al., 2024), involve a step-by-step evolution of tables through a long chain of transformations. In each new step, both the original table and the newly generated sub-table must be processed by language models. Additionally, their iterative process is complex, unstable, and difficult to analyze theoretically. In contrast, our approach is general and comprehensive, avoiding the trivial overhead of sub-table extraction. Instead, it focuses on holistic reasoning while maintaining ideal efficiency.

On the first 100 examples of TabFact, we evaluate *TableMaster* against the representative baseline CHAIN-OF-TABLE (Wang et al., 2024) using GPT-4o, ensuring a fair comparison without self-consistency decoding.

Table 9: Token usage comparison on the TabFact subset using GPT-4o.

| Method | Prompt Tokens | Completion Tokens |
|---|---|---|
| Chain-of-Table | 13,209.6 | 914.2 |
| TableMaster (Ours) | 3,393.5 | 738.6 |

As shown in Table 9, *TableMaster* consumes substantially fewer tokens while maintaining strong performance. While our design may appear intricate, it integrates practical features such as fallbacks to early-exit or full-table reasoning, thereby avoiding unnecessary table transformations. In contrast, CHAIN-OF-TABLE continuously evolves the table without fallback safeguards, leading to much higher token usage. This limitation is not acknowledged in the original paper, whereas our framework explicitly incorporates token efficiency as a design principle.

## J DETAILED ALGORITHM OF TABLE-OF-FOCUS RE-CONSTRUCTION

Here, we provide a detailed description of the Table-of-Focus Re-Construction algorithm, as shown in Algorithm 1.

---

**Algorithm 1** Algorithm of Table-of-Focus Re-Construction

---
**Require:** $\mathbb{T}$: The original table
**Require:** $Q$: The question
**Require:** $R$: Selected rows
**Require:** $C^0$: Initially selected columns
**Require:** $\mathbb{C}$: Ranked candidate column indices
**Ensure:** $\mathbb{T}^F$: Final table-of-focus
**Ensure:** $C$: Updated selected columns
 1: Initialize $C^{candidate} \leftarrow \{c \in \mathbb{C} \mid c \notin C^0\}$
 2: Initialize $C \leftarrow \text{Copy}(C^0)$
 3: **while true do**
 4:     $\mathbb{T}^F \leftarrow \text{extractTable}(\mathbb{T}, R, C)$
 5:     $E \leftarrow \text{estimateInformation}(\mathbb{T}^F, Q)$
 6:     **if** $E$ **or** $\text{len}(C^{candidate}) = \emptyset$ **then**
 7:        **break**
 8:     **else**
 9:        $c \leftarrow \text{popFront}(C^{candidate})$                          {Select the next candidate column}
10:        $C \leftarrow C \cup \{c\}$
11:     **end if**
12: **end while**
13: **return** $\mathbb{T}^F, C$

---

The Table-of-Focus Re-Construction Algorithm iteratively refines a table by selecting relevant columns to form the final table-of-focus $\mathbb{T}^F$. It starts by initializing the set of candidate columns $C^{candidate}$ that are not part of the initially selected columns $C^0$, and copies $C^0$ to initialize $C$. In each iteration, it extracts a sub-table $\mathbb{T}^F$ using the current selected columns and estimates whether the extracted sub-table contains sufficient information to answer the given question $Q$. If the information is sufficient $E = \text{True}$ or no more candidate columns remain, the process terminates. Otherwise, the next ranked candidate column is selected and added to $C$, repeating the process. The algorithm ultimately returns the refined table $\mathbb{T}^F$ and the updated set of selected columns, ensuring an efficient and structured approach to dynamically refining a table while balancing relevance and minimal table size.

## K    ANALYSIS OF TABLE-OF-FOCUS RE-CONSTRUCTION

Table 10: Column Selection Statistics Before and After Table-of-Focus Re-Construction for TabFact and WikiTQ.

| Dataset | Initial Columns | | Final Columns | | Added Columns | |
|---|---|---|---|---|---|---|
| | Number (#) | Percentage (%) | Number (#) | Percentage (%) | Number (#) | Percentage (%) |
| TabFact | 2.44 | 40.74 | 3.34 | 54.64 | 0.90 | 13.91 |
| WikiTQ | 2.87 | 47.67 | 4.72 | 75.91 | 1.85 | 28.23 |

Table 10 presents Column Selection Statistics before and after Table-of-Focus Re-Construction for two datasets: TabFact and WikiTQ. The table measures how many columns were initially selected, how many remained after refinement, and how many were newly added during the reconstruction process.

The table is structured into three main sections: Initial Columns, Final Columns, and Added Columns. Each section includes two metrics: the number of columns and the percentage of total columns in the dataset. The Initial Columns represent the starting number of columns before any refinement. The Final Columns show the number of columns retained after the reconstruction process. The Added Columns indicate the number of additional columns incorporated to enhance table comprehension.

For the TabFact dataset, the number of Initial Columns is 2.44, covering 40.74% of the table's total columns. After the reconstruction process, the Final Columns increase to 3.34, covering 54.64%. This means that 0.90 additional columns were introduced averagely, which accounts for 13.91% of the total columns. For the WikiTQ dataset, the pattern is similar but with higher values. The Initial Columns start at 2.87, representing 47.67% of the total table. After reconstruction, the Final Columns expand to 4.72, covering 75.91% of the table's total structure. This increase results from 1.85 additional columns, which make up 28.23% of the total columns.

Overall, this mechanism has been proven to be effective while remaining lightweight. The table demonstrates that Table-of-Focus Re-Construction slightly increases the number of selected columns, with a more pronounced effect in the WikiTQ dataset compared to TabFact. This suggests that WikiTQ tables require a greater degree of expansion to ensure adequate information coverage, whereas TabFact tables undergo a more moderate refinement process.

## L  ANALYSIS OF ADAPTIVE REASONING

Table 11: Performance of Different Reasoning Methods Across Base LLMs

| Base LLM | Method | Calculation Required #2692 | No Calculation Required #1652 | Overall #4344 |
|---|---|---|---|---|
| gpt-4o | Textual Reasoning | 81.17 | 88.56 | 83.98 |
| | Symbolic Reasoning | 74.59 | 74.70 | 74.63 |
| | Text-Guided Symbolic Reasoning | 76.49 | 77.36 | 76.82 |
| gpt-4o-mini | Textual Reasoning | 67.50 | 81.90 | 72.97 |
| | Symbolic Reasoning | 61.55 | 62.29 | 61.83 |
| | Text-Guided Symbolic Reasoning | 67.24 | 71.43 | 68.83 |
| gpt-3.5-turbo | Textual Reasoning | 52.27 | 72.40 | 59.92 |
| | Symbolic Reasoning | 43.28 | 61.80 | 50.32 |
| | Text-Guided Symbolic Reasoning | 59.10 | 66.65 | 61.97 |

We consider adaptive reasoning a key component in table understanding. Concurrent work, such as Abhyankar et al. (2025), also explores this direction.

Table 11 compares different reasoning methods—textual reasoning, symbolic reasoning, and text-guided symbolic reasoning—across various LLMs under calculation-required and no-calculation-required scenarios. This experiment is conducted using *gpt-4o-mini* on the WikiTQ dataset.

For *gpt-4o*, textual reasoning achieves the highest accuracy (83.98% overall), excelling in both calculation-required (81.17%) and no-calculation-required (88.56%) cases. Symbolic reasoning performs worse (74.63% overall), while text-guided symbolic reasoning offers slight improvements (76.82%). For *gpt-4o-mini*, a similar trend is observed, with textual reasoning maintaining the highest accuracy (72.97% overall), followed by text-guided symbolic reasoning (68.83%), and symbolic reasoning performing the worst (61.83%). For *gpt-3.5-turbo*, performance drops significantly, with textual reasoning at 59.92%, symbolic reasoning struggling at 50.32%, and text-guided symbolic reasoning achieving the best results (61.97%), indicating that symbolic guidance benefits weaker models.

Symbolic reasoning is consistently outperformed by textual reasoning, while text-guided symbolic reasoning surpasses textual reasoning only in *gpt-3.5-turbo* under calculation-required scenarios. One reason for this is that not all calculation-required questions necessarily benefit from symbolic reasoning; for simple calculations, textual reasoning is more effective. However, for complex calculation-required questions, text-guided symbolic reasoning is the preferred approach. This provides a key insight for prompt design of reasoning strategy assessment.

Overall, textual reasoning consistently outperforms symbolic reasoning across all models, while text-guided symbolic reasoning helps mitigate weaker numerical capabilities in smaller models. These results suggest that adaptive reasoning should prioritize textual approaches, incorporating symbolic methods selectively for numerical calculations in weaker models.

Table 12 compares the performance (accuracy %) and inference times of various reasoning methods, including chain of thought (CoT), program of thought (PoT), text-guided program of thought (TPoT), self-consistency, and adaptive reasoning. This experiment is conducted using *gpt-4o-mini* on the WikiTQ dataset.

Table 12: Performance and Inference Times for Different Methods

| Method | Accuracy (%) | Inference Times (#) |
|---|---|---|
| Chain of Thought | 72.97 | 1 |
| Program of Thought | 61.83 | 1 |
| Text-Guided Program of Thought | 68.83 | 1 |
| Self-Consistency (5 CoT) | 74.98 | 3 |
| Self-Consistency (5 PoT) | 63.97 | 3 |
| Mix Self-Consistency (3+3) | 76.70 | 6 |
| Mix Self-Consistency (5+5) | 77.46 | 10 |
| Self-Eval | 70.58 | 2 |
| Adaptive Reasoning (POT) | 71.18 | 1 |
| Adaptive Reasoning (TPOT) | 74.08 | 1 |
| Adaptive Reasoning (POT - Upper Bound) | 82.99 | 1 |
| Adaptive Reasoning (TPOT - Upper Bound) | 85.06 | 1 |

Among single-pass methods (1 inference), chain of thought achieves 72.97% accuracy, outperforming program of thought (61.83%) and text-guided program of thought (68.83%). This suggests that CoT is more effective than pure symbolic reasoning when only one inference is allowed.

Self-consistency methods, which perform multiple inferences to improve reliability, achieve better results. Five-shot CoT self-consistency reaches 74.98%, while five-shot PoT self-consistency lags behind at 63.97%. As introduced in (Liu et al., 2024b), mixed self-consistency (3 CoT + 3 PoT) and (5+5) further improve accuracy to 76.70% and 77.46%, respectively, at the cost of increased inference time (6 and 10 passes). Self-evaluation (self-eval) first performs CoT and PoT inferences (Prompt 26), then selects the better result, achieving 70.58% with 2 inferences.

Adaptive reasoning achieves competitive performance while maintaining single-pass efficiency. PoT-based adaptive reasoning reaches 71.18%, while TPOT-based adaptive reasoning, which combines textual and text-guided symbolic methods, improves to 74.08%. The upper-bound performance of these adaptive strategies (assuming perfect strategy selection) reaches 82.99% (PoT) and 85.06% (TPOT), significantly outperforming all other methods, highlighting the importance of textual guidance and strategy selection.

For the selection distribution between CoT and PoT (TPoT):

- **Self-eval:** 1,962 PoT and 2,382 CoT
- **Adaptive reasoning (PoT):** 1,590 PoT and 2,754 CoT
- **Adaptive reasoning (TPoT):** 1,590 PoT and 2,754 CoT
- **Adaptive reasoning (PoT - upper bound):** 435 PoT and 3,909 CoT
- **Adaptive reasoning (TPoT - upper bound):** 525 PoT and 3,819 CoT

These results suggest that language models should prioritize textual reasoning and reserve symbolic reasoning for more complex numerical calculations where it provides a clear advantage.

Overall, self-consistency enhances accuracy but requires multiple inferences, whereas adaptive reasoning effectively balances accuracy and efficiency. To further improve strategy assessment, we will explore ways to approach this upper bound in future work. This demonstrates that well-designed adaptive reasoning strategies can rival more computationally expensive self-consistency methods while maintaining efficiency.

## M    INFORMATION MISSING AND TABLE REASONING WITH FULL TABLE

As discussed in our limitations, the table-of-focus process may sometimes lead to the loss or omission of key relevant information. This issue is inevitable when attempting to locate specific data. If no relevant data exists within the selected portion, the reasoning result will naturally be incorrect.

Table 13: Performance comparison of reasoning with and without the full table on WikiTQ and TabFact.

| Method | WikiTQ | TabFact |
|---|---|---|
| PoTable (Previous SOTA) (Mao et al., 2024) | 64.73 | 88.93 |
| TableMaster w/ Full Table in Reasoning | 78.13 | 90.12 |
| TableMaster w/o Full Table in Reasoning | 77.23 (-0.90) | 89.58 (-0.54) |

In our experiments, we found that when using the table-of-focus and its verbalized representation for reasoning, 265 out of 4,344 questions in the WikiTQ dataset had no available answers. This led to a performance drop, as the language model responded with an inability to provide an answer. To address this, we replaced the table-of-focus with the original full table, combined with verbalized table-of-focus as input in those questions. The performance under this adjustment is shown in Table 13, reaching 77.23% in WikiTQ. When we directly replaced the table-of-focus with the full table for all questions in WikiTQ, the performance increased to 78.13%, resulting in a slight improvement of 0.9%. Two results are similar.

We believe this approach does not contradict previous steps such as structure extraction and table-of-focus selection. These steps remain valuable, as the extracted target data is retained in the verbalized table, where the information density is higher and semantic context is richer. The language model prioritizes this high-density information, and if it is insufficient, it can then reference the global information from the full table. This demonstrates the complementary nature of the full table and the verbalized table-of-focus. From an efficiency perspective, it is preferable to use the sub-table for reasoning initially and only switch to the full table when necessary.

To highlight the performance of *TableMaster*, we report the best scores of 78.13 and 90.12 in the main results table. Regardless, our method consistently outperforms the previous state-of-the-art, PoTable (Mao et al., 2024), on both WikiTQ and TabFact.

# N  CASE STUDY

## N.1  CASE STUDY OF TABLE VERBALIZATION

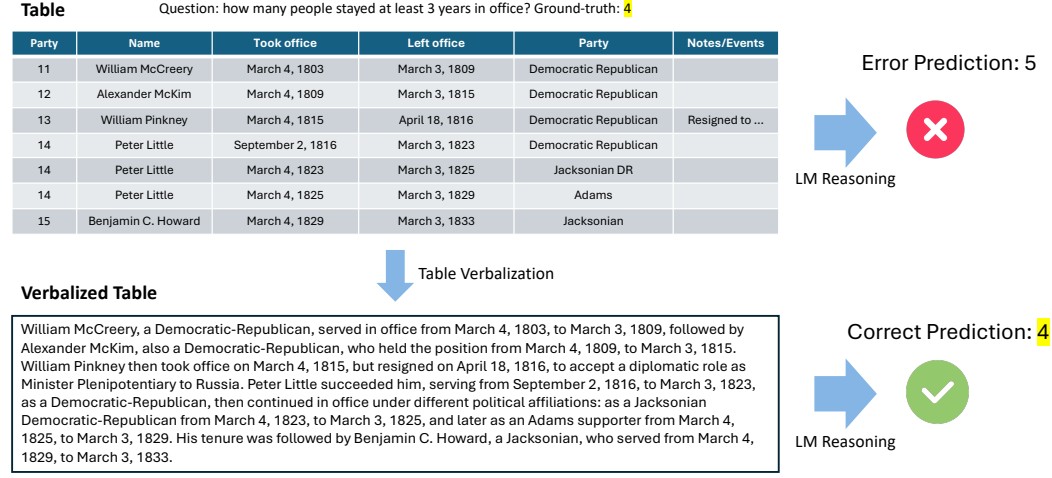

Figure 8: Case study on the impact of table verbalization. The data is from the WikiTQ dataset.

Table verbalization brings a slight overall improvement in table understanding and is particularly effective in cases where deeper comprehension of the table's context is required to answer questions accurately.

Figure 8 presents a case study on the impact of table verbalization in helping language models reason about structured data. The setup includes a table listing U.S. congressional representatives, their terms in office, political affiliations, and notable events. The question posed is: *How many people stayed at least 3 years in office?*, with a ground-truth answer of 4.

When the table is input directly, the model incorrectly predicts 5, as it mistakenly counts rows rather than identifying unique individuals. This suggests that the model relies on simple row counting instead of truly understanding the data. However, with the verbalized table, the model accurately interprets the descriptions, grasps the actual meaning, and correctly answers with 4.

## N.2   CASE STUDY OF *TableMaster*

As shown in Figure 9, we present a case study of *TableMaster* to illustrate its detailed workflow in answering the question: "Total wins by Belgian riders?" with a ground-truth answer of 7. The process begins with structure extraction, identifying key columns such as Rider, Country, and Wins. Next, column lookup selects relevant data, and row lookup filters the rows containing Belgian riders. SQL generation and execution retrieve only the relevant records where Country = Belgium.

The refined table is then constructed into a table-of-focus, keeping only the necessary columns. Table verbalization converts structured data into a description to enrich semantic context, providing insights into the number of wins for each Belgian rider. A textual guidance module generates a structured step-by-step explanation of the counting process, ensuring clarity in symbolic reasoning. The reasoning and execution phase involves symbolic reasoning (Program of Thought), where a Python snippet correctly extracts and sums the wins, leading to the correct prediction of 7.

This case study highlights *TableMaster* 's ability to accurately extract, process, and reason over structured data, demonstrating its effectiveness in table-based question answering with a combination of structured queries, reasoning steps, and code execution.

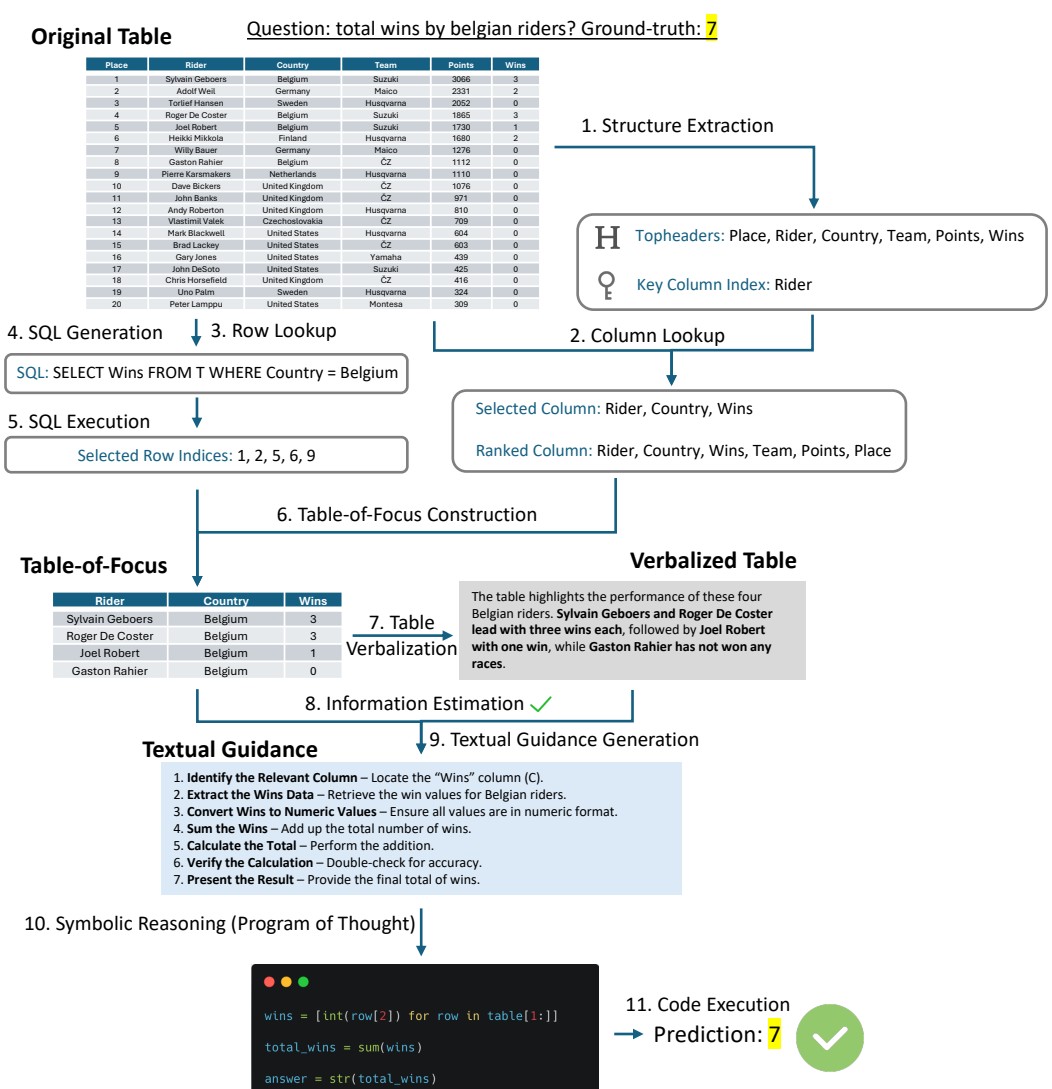

Figure 9: Case study of *TableMaster*. The data is from the WikiTQ dataset.

## O PROMPT DESIGN IN *TableMaster*

**Prompt for TableMaster – Structure Extraction**

---

*## Objective*
*You are provided with a text representation of a table in string format, detailing the content of each cell.*
*Your task is to identify and extract the Top Header and Key Column of the table.*

*## Table Definition*
*The table is represented by cell-value pairs, where each pair consists of a cell address and a value of the content in that cell, separated by a comma (e.g., 'A1,Year').*
*Multiple cells are separated by '|' (e.g., 'A1,Year|A2,Profit').*
*Cells may contain empty values, represented as 'A1,|A2,Profit'.*

*## Table*
*{table}*

*## Instructions*
*1. Top Header: The section at the top of the table, often spanning multiple columns horizontally, that describes the primary information presented in the table.*
*2. Key Column: A column where the values best represent the subject or key identifier for each row in the table, typically containing row labels or keys (e.g., year, date, number, name, etc.).*
*3. You should extract the top headers with address and value, like ['A1,Year', 'A2,Profit', ...].*
*4. key_column_index should be like 'A' or 'B' ...*
*5. The key column should contain meaningful values instead of id.*

*## Response Format*
*The response should be in JSON format:*
*```json*
*{{*
*  "topheaders": ["address1,header1", "address2,header2", ...],*
*  "key_column_index": "column1",*
*}}*
*```*

---

Figure 10: Prompt for structure extraction in *TableMaster*. Blue text indicates placeholders for variables within the prompt. The prompt guides the language model in extracting the table's structure.

**Prompt for TableMaster – Column Ranking**

```
## Objective
You are provided with information of a table and a question related to the table.
Your task is to rank the column indices based on the relevance to the question.

## Table Definition
The table is represented by cell-value pairs, where each pair consists of a cell address and a value of the content in that cell, separated
by a comma (e.g., 'A1,Year').
Multiple cells are separated by '|' (e.g., 'A1,Year|A2,Profit').
Cells may contain empty values, represented as 'A1,|A2,Profit'.

## Table Information
Table:
{table}

Top Headers: {topheaders}

## Question
{question}

## Instructions
1. The column indices must only contain letters, like ['A', 'B', 'C', ...].
2. You should first rank all the column indices based on the relevance to the question.
3. Your output should contain all the column indices.

## Response Format
The response should be in JSON format:
```json
{{
    "ranked_column_indices": ["column indexA", "column indexB", ...]
}}
```
```

Figure 11: Prompt for column ranking in *TableMaster*. Blue text indicates placeholders for variables within the prompt. The prompt guides the language model to rank the priority of all columns based on the given table, top headers, and related question.

**Prompt for TableMaster – Column Lookup**

```
## Objective
You are provided with information of a table and a question related to the table.
Your task is to lookup the column indices that are needed to answer the question based on the table.

## Table Definition
The table is represented by cell-value pairs, where each pair consists of a cell address and a value of the content in that cell, separated
by a comma (e.g., 'A1,Year').
Multiple cells are separated by '|' (e.g., 'A1,Year|A2,Profit').
Cells may contain empty values, represented as 'A1,|A2,Profit'.

## Table Information
Table:
{table}

Top Headers: {topheaders}

## Question
{question}

## Instructions
1. The column indices must only contain letters, like ['A', 'B', 'C', ...].
2. Your output of the column indices should not any contain number, like ['A1', 'B2', 'C1', ...].
3. Your output of the column indices should not contain the column name.
4. You should select the column that are relevant and necessary to answer the question.

## Response Format
The response should be in JSON format:
```json
{{
    "selected_column_indices": ["column indexA", "column indexB", ...]
}}
```
```

Figure 12: Prompt for column lookup in *TableMaster*. Blue text indicates placeholders for variables within the prompt. The prompt guides the language model to select relevant columns based on the given table, top headers, and related question.

**Prompt for TableMaster – SQL Generation for Row Lookup**

```
## Objective
You are provided with information of a table and a question related to the table.
Your task is to generate a SQL query that can be used to find the rows that answer the question.

## Table Information
Part of Table:
{table}

## Question
{question}

## Instructions
1. The SQL query must be in the format of `SELECT XXX, ... FROM Table WHERE XXX ...`,
where Table is the table name, XXX is the column name, and WHERE... is the criteria.
2. If the information is not enough to answer the question, you should return a sql to select all rows.
3. Do not give complex sql query, just simple query to select rows.
4. Use this SQL query only to select relevant rows, not for getting the final answer.

## Response Format
Provide the response in the following JSON format:
```json
{{
    "sql": "SELECT XXX, ... FROM Table WHERE XXX ..."
}}
```
```

Figure 13: Prompt for SQL generation for row lookup in *TableMaster*. Blue text indicates placeholders for variables within the prompt. The prompt guides the language model to generate SQL for selecting relevant rows based on the given table and related question.

**Prompt for TableMaster – Table Verbalization**

```
## Objective
You are provided with a table in string format.
Your task is to convert the table into a detailed text description.

## Table
{table}

## Instructions
1. Provide a detailed description of the table, covering all rows and columns.
2. Include every detail and numerical value without omitting or summarizing.
3. Use external knowledge only to enhance clarity, while staying faithful to the table's content.
4. If the table only contains headers and no rows, it should be described as an empty table.

Now, please provide the verbalized description of the table:
```

Figure 14: Prompt for table verbalization in *TableMaster*. Blue text indicates placeholders for variables within the prompt. The prompt guides the language model to verbalize the given table by adding detailed descriptions and additional knowledge about the table.

**Prompt for TableMaster – Information Estimation**

```
## Objective
You are provided with information from a table and a question related to the table.
Your task is to estimate whether the current information of the table can answer the question.

## Table Information
Top Headers: {topheader_info}
Table Content:
{table}

## Question
{question}

## Response Format
The response should be in JSON format:
```json
{{
    "results": True of False
}}
```
```

Figure 15: Prompt for information estimation in *TableMaster*. Blue text indicates placeholders for variables within the prompt. The prompt guides the language model to evaluate the given table's content and determine whether it contains sufficient information to answer the provided question

**Prompt for TableMaster – Reasoning Strategy Assessment**

```
## Objective
You are provided with a table and a question related to the table.
Your task is to assess whether answering this question needs mathematical calculation.

## Table
{table}

## Question
{question}

## Instructions
1. If the question can be directly answered using the information in the table, you should respond with `False`.
2. If the question involves counting something, you should respond with `True`.
3. If the question requires calculations based on the data in the table, you should respond with `True`.

## Response Format
The response should be in JSON format:
```json
{{
    "results": True of False
}}
```
```

Figure 16: Prompt for reasoning strategy assessment in *TableMaster*. Blue text indicates placeholders for variables within the prompt. The prompt guides the language model to evaluate whether answering the given question requires direct information retrieval, counting, or mathematical calculations based on the table's content. The response determines the subsequent reasoning strategy.

**Prompt for TableMaster – Textual Reasoning**

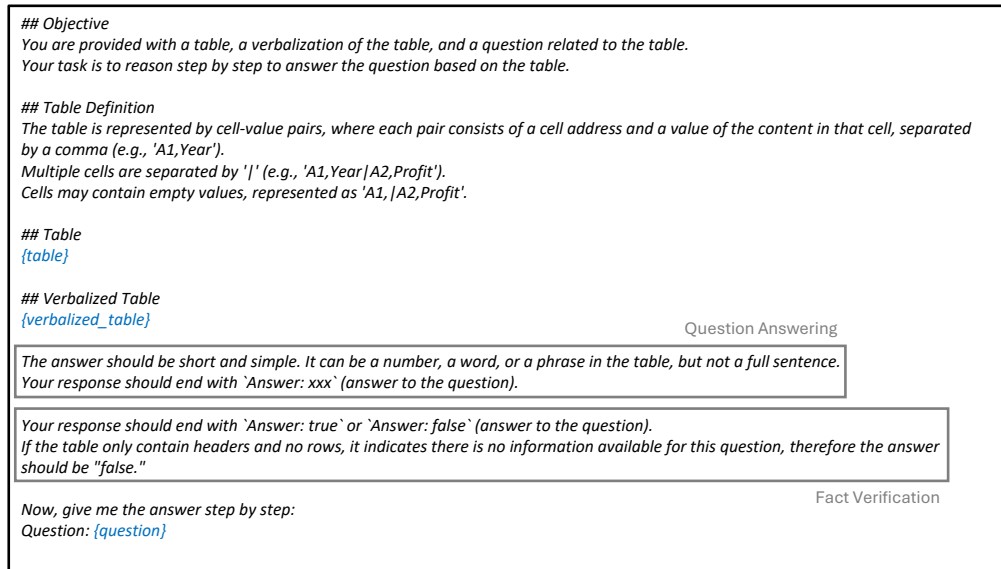

Figure 17: Prompt for textual reasoning in *TableMaster*. Blue text represents placeholders for variables within the prompt, while the grey region indicates optional sections to adapt the prompt for question-answering or fact-verification tasks. The prompt guides the language model to answer the question step by step.

**Prompt for TableMaster – Textual Guidance Generation**

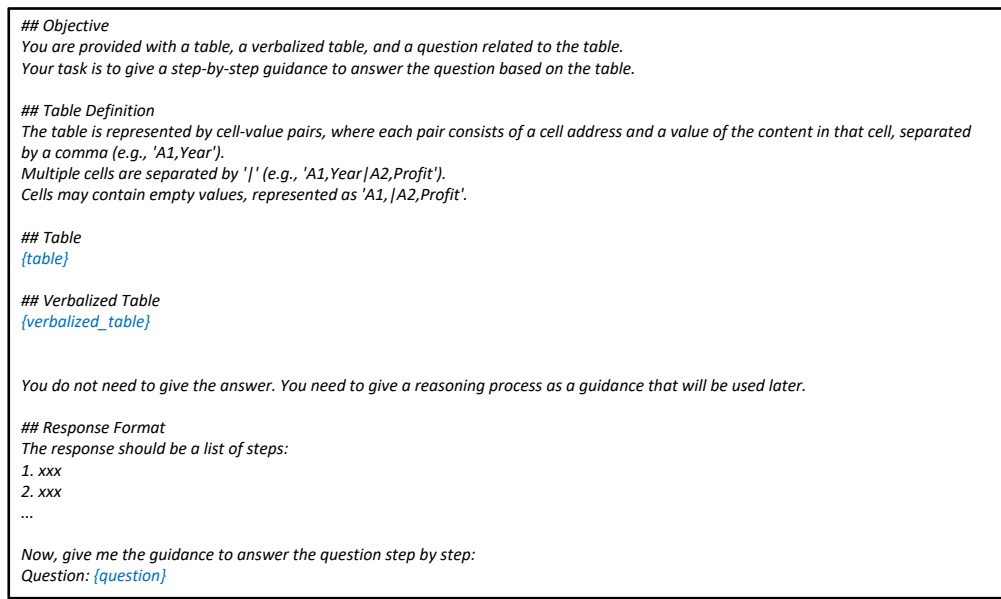

Figure 18: Prompt for textual guidance generation in *TableMaster*. Blue text indicates placeholders for variables within the prompt. The prompt guides the language model to generate textual guidance that can be utilized for subsequent symbolic reasoning.

**Prompt for TableMaster – Symbolic Reasoning (Programming of Thought)**

*## Objective*
*You are provided with a table, a verbalized table, a guidance, and a question related to the table.*
*Your task is to generate Python code that answers the question using the table and the guidance as a guide.*

*## Table Definition*
*The table is represented by cell-value pairs, where each pair consists of a cell address and a value of the content in that cell, separated by a comma (e.g., 'A1,Year').*
*Multiple cells are separated by '|' (e.g., 'A1,Year|A2,Profit').*
*Cells may contain empty values, represented as 'A1,|A2,Profit'.*

*## Table*
*{table}*

*## Verbalized Table*
*{verbalized_table}*

*## Guidance*
*{textual_guidance}*

*## Question*
*{question}*

*## Instructions*
*1. The actual data of the table is stored in the variable `table` as a list of lists.*
*2. The result should be store in the variable `answer` as a string and do not need to print it.*
*3. You need to generate Python code within ```python``` code block.*

*Now, give me the executable python code to answer the question:*
*```python*
*table = {table_array}*

Figure 19: Prompt for symbolic reasoning in *TableMaster*. Blue text indicates placeholders for variables within the prompt. The prompt guides the language model to generate Python code to answer the question.

**Prompt for TableMaster – Answer Formatting**

*## Objective*
*You are provided with a process of text-guided reasoning with programming and a question related to the table.*
*Your task is to answer the question using the reasoning process.*

*## Table Definition*
*The table is represented by cell-value pairs, where each pair consists of a cell address and a value of the content in that cell, separated by a comma (e.g., 'A1,Year').*
*Multiple cells are separated by '|' (e.g., 'A1,Year|A2,Profit').*
*Cells may contain empty values, represented as 'A1,|A2,Profit'.*

*## Table*
*{table}*

*## Textual Reasoning Process*
*{textual_reasoning_process}*

*## Programmed Reasoning Process*
*{symbolic_reasoning_process}*

*The answer should be short and simple. It can be a number, a word, or a phrase in the table, but not a full sentence.*
*Your response should be in the format of `Answer: xxx` (answer to the question).*

*Question: {question}*
*Answer:*

Figure 20: Prompt for answer formatting in *TableMaster*. Blue text indicates placeholders for variables within the prompt. The prompt guides the language model to format the final answer based on the given table, question, and reasoning process.

## P  PROMPTS USED IN ANALYSIS EXPERIMENTS

**Prompt for Direct Baseline**

---

*## Objective*
*You are provided with a table and a question related to the table.*
*Your task is to answer the question directly based on the table.*

*## Table*
*{table}*

*## Question*
*{question}*

*The answer should be short and simple. It can be a number, a word, or a phrase in the table, but not a full sentence.*
*Now, answer the question directly:*
*Answer:*

---

Figure 21: Direct prompt for table understanding in analysis experiment. Blue text indicates place-holders for variables within the prompt. The prompt guides the language model to directly give the final answer based on the given table and question.

**Prompt for Chain of Thought Baseline**

---

*## Objective*
*You are provided with a table and a question related to the table.*
*Your task is to answer the question step by step based on the table.*

*## Table*
*{table}*

*## Question*
*{question}*

*The answer should be short and simple. It can be a number, a word, or a phrase in the table, but not a full sentence.*
*Your response should end with `Answer: xxx` (answer to the question).*
*Now, answer the question step by step:*

---

Figure 22: Chain of thought prompt for table understanding in analysis experiment. Blue text indicates placeholders for variables within the prompt. The prompt guides the language model to give the answer step by step based on the given table and question.

**Prompt for Program of Thought Baseline**

*## Objective*
*You are provided with a table and a question related to the table.*
*Your task is to answer the question based on the table by writing python code as a solution.*

*## Table*
*{table}*

*## Reasoning Instructions*
*1. You must use executable python code to solve the question.*
*2. The final answer should be variable named "answer" in the code.*
*3. Do not execute the code in the response.*
*4. The python code should be in the following format:*
*```python*
*# your code here*
*```*

*Now, answer the question by writing python code as a solution:*
*Question: {question}*
*```python*

Figure 23: Program of thought prompt for table understanding in analysis experiment. Blue text indicates placeholders for variables within the prompt. The prompt guides the language model to generate code to derive the answer based on the given table and question.

**Prompt for Verbalization Baseline**

*## Objective*
*You are provided with a table in string format.*
*Your task is to convert the table into a detailed text description.*

*## Table*
*{table}*

*## Instructions*
*1. Provide a comprehensive description of the table.*
*2. Include all details and numerical values from the table in your response.*
*3. Do not omit or summarize any information from the table.*
*4. You may use external knowledge to enhance your understanding of the table, but the response must remain faithful to the table's content.*

*Now, please provide the verbalized description of the table:*

Figure 24: Prompt for table verbalization in analysis experiment. Blue text indicates placeholders for variables within the prompt. The prompt guides the language model to verbalize a table to add detailed description.

**Prompt for Textual Guidance Generation**

*## Objective*
*You are provided with a table, and a question related to the table.*
*Your task is to give a step-by-step guidance to answer the question based on the table.*

*## Table Definition*
*The table is represented by cell-value pairs, where each pair consists of a cell address and a value of the content in that cell, separated by a comma (e.g., 'A1,Year').*
*Multiple cells are separated by '|' (e.g., 'A1,Year|A2,Profit').*
*Cells may contain empty values, represented as 'A1,|A2,Profit'.*

*## Table*
*{table}*

*You do not need to give the answer. You need to give a reasoning process as a guidance that will be used later.*
*Keep the reasoning process concise and clear.*
*Control the number of steps in the reasoning process in the range of 1-5.*

*## Response Format*
*The response should be a list of steps:*
*1. xxx*
*2. xxx*
*...*

*Now, give me the guidance to answer the question step by step:*
*Question: {question}*

Figure 25: Prompt for textual guidance generation in analysis experiment. Blue text indicates placeholders for variables within the prompt. The prompt guides the language model to generate textual guidance that used for symbolic reasoning.

**Prompt for Reasoning Strategy Evaluation**

*Question: {question}*

*Table: {table}*

*Method 1 Solution: {cot_prediction}*
*Method 1 Reasoning: {cot_reasoning}*

*Method 2 Solution: {pot_prediction}*
*Method 2 Reasoning: {pot_reasoning}*

*Please evaluate which method is better.*
*Respond in the following JSON format:*
*{{*
*   "better_method": 1 or 2*
*}}*

Figure 26: Prompt for reasoning strategy evaluation in analysis experiment. Blue text indicates placeholders for variables within the prompt. The prompt guides the language model to select the better reasoning process after table reasoning.

## Prompt for Reasoning Strategy Assessment in Adaptive Reasoning

*You are provided with a table and a question related to the table.*
*Your task is to assess whether answering this question needs mathematical calculation.*

*## Table*
*{table}*

*## Question*
*{question}*

*## Instructions*
*1. If the question can be easily answered using the information in the table, respond with False.*
*2. If the question involves comparison, respond with False.*
*2. When the question involves counting a substantial number (more than 5) of items or rows, respond with True.*
*3. If the question demands complex calculations or multi-step mathematical operations based on the table's data, the response should be True.*
*4. For simple arithmetic or small-scale counting that requires minimal computational effort, respond with False.*

*## Response Format*
*The response should be in JSON format:*
*```json*
*{{*
*    "need_calculation": true/false*
*}}*
*```*

Figure 27: Prompt for reasoning strategy evaluation in analysis experiment. Blue text indicates placeholders for variables within the prompt. The prompt guides the language model to select the better reasoning strategy before table reasoning.

## Prompt for Question Type Classification (Calculation Required)

*Table:*
*{table}*

*Question:*
*{question}*

*Determine whether a calculation is required to answer the question, or if the question can be directly answered using the information in the table.*

*Provide your response in the following JSON format:*
*{{*
*    "need_calculation": true/false*
*}}*

Figure 28: Prompt for classifying a question type based on whether calculation is required in the analysis experiment. Blue text indicates placeholders for variables within the prompt.

**Prompt for Noised Table Generation**

*Given a table, you need to generate a new table by disrupting the content in the table.*

*Table:*
*{table}*

*Rules:*
*- Your goal is to make the content in each row within the same column follows a different format to increase diversity as much as possible.*
*- You cannot change the structure of the table.*
*- You cannot add or remove any rows or columns.*
*- You cannot modify the column names in the first row.*
*- You can only alter the format of the content in each cell, not the actual values.*
*- You should not make the content in each row within the same column in the same format as much as possible.*

*Format Change Examples:*
*- Change a number format from 123456 to 123,456.*
*- Change a date format from 2024-01-01 to 2024/01/01.*
*- Simplify or abbreviate text content.*

*Provide your new table in the following JSON format:*
*```json*
*{{*
*  "table": [[...], [...], [...]],*
*}}*
*```*

Figure 29: Prompt for generating noised tables in the analysis experiment. Blue text represents placeholders for variables within the prompt. The prompt instructs the language model to add noise by altering the cell content format based on a given table.

# Q NOTATION TABLE

Table 14 provides a comprehensive list of the notations used throughout this paper, along with their corresponding descriptions. This table serves as a quick reference to help readers better understand the concepts presented in our work.

Table 14: Notation used throughout the paper

| Notation | Description |
|---|---|
| *General* | |
| $Q$ | Given question or query |
| $A$ | Generated answer |
| $\mathbb{T}$ | Input table |
| $\mathbb{T}^W$ | Wild table before normalization |
| $\mathbb{T}^N$ | Normalized table |
| $\mathbb{T}^F$ | Table-of-Focus |
| $C_{i,j}$ | Cell in the $i$-th row and $j$-th column |
| $m, n$ | Number of rows and columns in the table |
| *Table Structure Understanding* | |
| $H$ | Set of top headers |
| $K$ | Key column serving as row identifier |
| $\mathbb{C}$ | Candidate column set |
| $C^0$ | Selected relevant columns |
| $R$ | Selected relevant rows |
| $k$ | Peek size for table processing |
| *Table Content Understanding* | |
| $T^{\mathbb{T}}$ | Verbalized table (natural language text) |
| $a', b'$ | Number of refined columns and rows after reconstruction |
| *Table Reasoning* | |
| $S$ | Selected reasoning strategy |
| $\mathcal{T}$ | Textual reasoning strategy |
| $\mathcal{S}$ | Symbolic reasoning strategy |
| $G$ | Textual reasoning guidance |
| $\mathcal{P}$ | Program executor (Python/SQL) |

## R    THE USE OF LARGE LANGUAGE MODELS (LLMS)

In this work, large language models (LLMs) were used *only to aid with writing and polishing the manuscript*. Specifically, LLMs were employed for grammar correction, phrasing suggestions, and improving readability. All research ideas, methodological contributions, theoretical analyses, and experiments were entirely conceived, designed, and executed by the authors without the involvement of LLMs. The authors take full responsibility for the scientific content of the paper.

