# OpenReview forum: "TableMaster: A Recipe to Advance Table Understanding with Language Models"
_ICLR.cc/2026/Conference — ICLR 2026 Poster_

### Official Review · Reviewer_oMBW · 2025-10-28

**Soundness:** 2
**Presentation:** 2
**Contribution:** 2
**Rating:** 4
**Confidence:** 3

**Summary:**

This paper proposes TableMaster, a framework designed to enhance language models'  table understanding capabilities. The paper identifies several challenges: difficulty in locating target data, semantic deficiency in tables, numerical inaccuracies in textual reasoning, and semantic inflexibility in symbolic reasoning. To address these, TableMaster integrates multiple solutions including table-of-focus construction, table verbalization, program-aided reasoning, and adaptive reasoning that dynamically selects between textual and symbolic approaches. Extensive experiments on WikiTQ, TabFact, and FetaQA datasets show that TableMaster achieves good performance.

**Strengths:**

1. The paper provides a structured analysis of four fundamental challenges in table understanding, with each challenge directly linked to a targeted solution.

2. TableMaster integrates multiple techniques (table-of-focus, verbalization, adaptive reasoning) into a pipeline.

3. The paper conducts extensive experiments across diverse datasets and LLMs.

**Weaknesses:**

1. The core contributions of the paper are primarily engineering-focused. The paper lacks novel advancements in LM architecture or reasoning mechanisms specific to table understanding.

2. Experiments are concentrated on clean, structured tables from specific domains. The framework's performance on real-world noisy tables, hierarchical tables remains insufficiently explored, for example, the BIRD dataset.

3. The dynamic strategy selection relies on LM judgment without robust error handling.

4. While efficiency is discussed, no actual latency measurements or comparison with simpler baselines are provided. The multi-step process (verbalization, reconstruction, adaptive reasoning) likely introduces significant inference time overhead.

**Questions:**

please refer to the weakness

---

> ### Author Response · Authors · 2025-11-26
>
> Thank you for the reviewer’s constructive and insightful comments. We sincerely appreciate the feedback, which has helped us significantly strengthen the clarity and completeness of the paper. Below we address each concern in detail.
>
>
> ---
>
> ## **[W2] Evaluation on Noisy / Hierarchical / Real-World Tables**
>
>
> Thank you for the insightful comment. Although the main paper focuses on standard clean table benchmarks, we have already evaluated TableMaster on hierarchical and more complex real-world tables. Specifically, in Appendix F.2: Evaluation on Hierarchical Tables with the HiTab Dataset, we show that TableMaster significantly improves performance over Chain-of-Table on HiTab—a benchmark explicitly designed for hierarchical and semi-structured tables. These results confirm that TableMaster remains effective beyond clean relational tables, and we will highlight this more clearly in the revision.
>
> | Method                                           | Accuracy |
> |--------------------------------------------------|----------|
> | MultiCoT (Chain-of-Table in multi-table version)                | 64.0     |
> | MultiCoT (optimized prompt)                      | 70.0     |
> | MultiCoT (optimized prompt + verbalized table)   | 73.5     |
> | **TableMaster**                                   | **74.2** |
>
>
>
>
> ---
>
> ## **[W3] Dynamic Strategy Selection and Error Handling**
>
> We appreciate the reviewer’s concern. As analyzed in Appendix L: Analysis of Adaptive Reasoning, the current design intentionally prioritizes one-shot efficiency, trading a small amount of accuracy for substantial speed improvements by avoiding multi-round self-consistency or program voting.
> While more sophisticated error-handling mechanisms (e.g., multi-step verification) could further improve accuracy, they would significantly increase inference overhead, undermining one of the core goals of TableMaster. We will clarify this design trade-off in the revision and discuss how additional error handling can be optionally integrated when efficiency is less critical.
>
> ---
>
> ## **[W4] Latency and Efficiency Evaluation**
>
>
> Thank you for pointing this out. In the current version, we report token usage rather than explicit latency because in LLM-based systems, prompt length is a strong proxy for real inference time. As shown in Table 9 (reproduced below), TableMaster reduces prompt token usage by over 4× compared to Chain-of-Table, while also improving accuracy:
>
> | Method             | Prompt Tokens | Completion Tokens |
> |--------------------|---------------|-------------------|
> | Chain-of-Table     | 13,209.6      | 914.2             |
> | **TableMaster (Ours)** | **3,393.5**      | **738.6**           |
>
> This demonstrates that TableMaster is not only more accurate but also substantially more efficient than baselines relying on large multi-step sub-table construction pipelines. We will make this connection more explicit and add a short note explaining the equivalence between token cost and latency measurements in LLM inference.
>
> ---

---

> ### Author Response · Authors · 2025-11-26
>
> ## **[W1] Novelty Clarification and Differentiation from Prior Work**
>
>
> Below we clarify the novelty of TableMaster and its position relative to prior work through three components: **Detailed Comparison**, **Motivation and Contribution**, and **Overview of Novelty & Meaningful Departure from Prior Work**.
>
> ---
>
> ### **1. Detailed Comparison**
>
> We compare TableMaster with prior studies that adapt LLMs for table understanding **without fine-tuning**, which is most relevant to our setup. Our framework is motivated by four core challenges identified in Section 3, while existing works typically address only one of them.
>
> ---
>
> #### **1.1 Difficulty in Locating Target Data**
>
> These methods primarily focus on sub-table construction, decomposition, or context reduction:
> - DATER [4]
> - TabSQLify [5]
> - ReAcTable [6]
> - TAP4LLM [7]
> - Tree-of-Table [8]
> - Chain-of-Table [9]
>
> **Our findings:**
> Despite their sophisticated designs, we observe that complex sub-table construction methods often perform poorly in practice and result in high computational cost—especially Tree-of-Table and Chain-of-Table. In contrast, our experiments show that the most reliable approach is **LLM-based column selection combined with SQL-based row filtering**, which achieves better performance and efficiency.
>
> In addition, prior works rarely address the **missing-information issue** in sub-table extraction. TableMaster explicitly tackles this through a **Fallback Strategy** (Appendix J, M), where we reconstruct incomplete subtables to avoid information loss—an important issue overlooked by earlier approaches.
>
> ---
>
> #### **1.2 Table Semantic Deficiency**
>
> Few works directly tackle semantic enrichment of tables.
> Some early ideas (table-to-text verbalization) suggest potential benefits, but they are not systematically integrated into reasoning pipelines.
>
> In TableMaster, we show that **structured verbalization enriches table semantics** and consistently improves reasoning quality, addressing semantic sparsity that existing methods leave unresolved.
>
> ---
>
> #### **1.3 Numerical Inaccuracy in Textual Reasoning**
>
> These methods introduce Python/SQL execution to mitigate numerical mistakes:
> - BINDER [1]
> - LEVER [2]
> - PoTable [3]
> - MIX-SC [10]
> - SpreadsheetEncoder [11]
>
> While symbolic execution is useful, prior works implicitly assume it is always preferable.
> Our analysis reveals that **textual reasoning is more effective for most table queries**, and symbolic execution should be used selectively, primarily in calculation-intensive scenarios. TableMaster introduces **Adaptive Reasoning**, which dynamically selects between textual and symbolic reasoning pathways for stronger and more reliable performance.
>
> ---
>
> #### **1.4 Semantic Inflexibility in Symbolic Reasoning**
>
> - MIX-SC [10]
>
> MIX-SC alleviates symbolic rigidity through 10-way self-consistency (5 textual + 5 symbolic), but at the cost of substantial latency. It also does not analyze why textual reasoning helps.
>
> TableMaster provides a **single-shot, efficient** alternative via Adaptive Reasoning, matching the benefits of self-consistency without its computational overhead, and explaining *why* textual reasoning works better in many cases.
>
> ---
> **References:**
>
> [1] Z. Cheng et al., “BINDER: Binding Language Models in Symbolic Reasoning Framework,” 2023.
> [2] A. Ni et al., “LEVER: Learning to Verify Language-to-Code Generation with Execution,” ICML, 2023.
> [3] Q. Mao et al., “PoTable: Programming Standardly on Table-based Reasoning Like a Human Analyst,” 2024.
> [4] Y. Ye et al., “DATER: Large Language Models are Versatile Decomposers for Table-based Reasoning,” 2023.
> [5] M. H. Nahid and D. Rafiei, “TabSQLify: Enhancing Reasoning Capabilities of LLMs Through Table Decomposition,” NAACL, 2024.
> [6] Z. Zhang et al., “ReAcTable: ReAct-style Table Reasoning via Intermediate Tables and Code Execution,” 2023.
> [7] J. Sui et al., “TAP4LLM: Table Prompting Toolbox for Large Language Models,” 2024.
> [8] Z. Ji et al., “Tree-of-Table: Hierarchical Sub-Table Construction for Table Question Answering,” 2024.
> [9] Z. Wang et al., “Chain-of-Table: Evolving Tables in the Reasoning Chain for Table Understanding,” 2024.
> [10] Q. Liu et al., “Rethinking Tabular Data Understanding with Large Language Models,” 2024.
> [11] W. Dong et al., “SpreadsheetEncoder: Interpreting Tabular Data within Spreadsheet Environments,” 2024.
>
>
> ---

---

> ### Author Response · Authors · 2025-11-26
>
> ## **[W1] Novelty Clarification and Differentiation from Prior Work (Cont.)**
>
> ### **2. Motivation and Contribution**
>
> LLMs struggle with tables due to structural complexity, semantic sparsity, and mixed reasoning demands. Existing works typically propose isolated improvements—better prompts, specialized retrievers, symbolic reasoning modules—but lack a **generalizable and unified framework**.
>
> TableMaster provides a **systematic, extensible recipe** that organizes:
> - table representation and semantic enrichment,
> - selective sub-table construction,
> - dynamic reasoning mode selection,
> - program execution,
> - fallback and reconstruction mechanisms,
>
> into a **cohesive and principled framework**.
> To our knowledge, TableMaster is the **first challenge-driven, holistic study** that evaluates, analyzes, and organizes LLM-based table reasoning into a unified design space.
>
> ---
>
> ### **3. Overview of Novelty and a Meaningful Departure from Prior Work**
>
> Although TableMaster uses components that have appeared independently in earlier works, it departs from them in several important ways:
>
> ---
>
> #### **3.1 A Systematic and Unified Recipe**
>
> Prior works focus on isolated improvements (e.g., a better retrieval scheme, a single reasoning pipeline, or a symbolic executor). TableMaster provides a **unified and extensible recipe** grounded in four core challenges, assembling fragmented insights into a coherent framework with broader generalization and transferability.
>
> ---
>
> #### **3.2 Mechanisms Beyond Simple Integration**
>
> We introduce new mechanisms that address practical gaps in earlier methods:
>
> - **Adaptive Reasoning (Appendix L):**
>   Dynamic switching between textual and symbolic reasoning without expensive self-consistency.
>   Prior works assume static reasoning or rely heavily on ensembles.
>
> - **Fallback Strategy in Table-of-Focus (Appendix J, M):**
>   A novel reconstruction mechanism that corrects incomplete subtables—an issue rarely handled explicitly in retrieval-based systems.
>
> ---
>
> #### **3.3 Providing Fundamental Insights Missing from Previous Work**
>
> Beyond algorithmic integration, TableMaster offers **deeper, empirical insights** that prior studies have not surfaced, including:
> - Why textual reasoning often outperforms symbolic reasoning and how to combine them,
> - Why complex sub-table extraction techniques fail in practice,
> - How semantic enrichment directly influences table reasoning quality,
> - How missing-information risks propagate through table reasoning pipelines,
> - What is the most efficient and effective way to make LLMs understand table better in each stage.
>
> These insights reshape how table reasoning should be framed for LLMs and provide conceptual guidance that earlier work lacks.
>
> ---
>
> #### **3.4 A Practical and Scalable Foundation**
>
> TableMaster is not an incremental extension; it represents a meaningful shift toward robust, efficient, and general-purpose table reasoning, offering a practical foundation for future research. Moreover, our conclusions are supported by extensive experiments across diverse datasets, reasoning types, and LLM backbones. Conducting these evaluations required substantial computational effort, further underscoring the depth, reliability, and scalability of our findings.
>
>
> ---
>
> Thank you again for your valuable feedback. We believe these updates strengthen the paper and adequately address the reviewer’s concerns.
>
> **Sincerely,**
> Authors of *TableMaster*

---

### Official Review · Reviewer_sJMb · 2025-11-01

**Soundness:** 2
**Presentation:** 2
**Contribution:** 1
**Rating:** 4
**Confidence:** 3

**Summary:**

This paper addresses table understanding with language models by identifying four key challenges: (i) difficulty in locating target data, (ii) deficiency of table semantics, (iii) numerical inaccuracy in textual reasoning, and (iv) semantic inflexibility in symbolic reasoning. The authors propose TableMaster, a comprehensive framework that integrates multiple solutions including table-of-focus construction, table verbalization, and adaptive reasoning that dynamically switches between textual and symbolic approaches. The method is evaluated on WikiTQ, TabFact, and FetaQA datasets, showing improvements over existing baselines.

**Strengths:**

- The paper provides a thorough empirical analysis of challenges in table understanding, with systematic experiments examining the impact of table size, verbalization, and different reasoning approaches.
- TableMaster achieves notable improvements across various large-scale LLMs (GPT-3.5-turbo, GPT-4o-mini, LLaMA-3 70B).

**Weaknesses:**

- While the integration is well-executed, most individual components (sub-table extraction, table verbalization, program-aided reasoning) have been proposed in prior literature. The novelty primarily lies in their combination rather than in introducing fundamentally new techniques.
- The section 3-4 can be condensed to leave more space for experiment and analysis. Currently, most the results are in the appendix.
- The evaluation focuses exclusively on large-scale models. It remains unclear how TableMaster performs on smaller (7–8B) models or what minimal model capabilities are required for it to function effectively.

**Questions:**

1. What are the minimum model capabilities required for TableMaster? Have you tested on 7-13B parameter models? At what model size does the framework start to break down?
2. Given that each component is well-established, what specific insights or contributions does TableMaster provide beyond engineering integration?

---

> ### Author Response · Authors · 2025-11-26
>
> Thank you for the reviewer’s constructive and insightful comments. We sincerely appreciate the feedback, which has helped us significantly strengthen the clarity and completeness of the paper. Below we address each concern in detail.
>
> ---
>
> ## **[W2] Section 3–4 Condensation and Experiment Relocation**
>
> Thank you for the suggestion. Our work involves a large amount of experimentation, and we aimed to provide as many insights and results as possible to benefit the community. This is why several analyses are currently placed in the appendix. In the revision, we will condense Sections 3–4 and reorganize the presentation so that more key results can be moved into the main paper. We hope that, with the additional space in the camera-ready version, we can present even more of these findings directly in the main text.
>
>
> ---
>
> ## **[W3 & Q1] Evaluation on Smaller Models and Minimum Capability Requirements**
>
> Thank you for raising this important question regarding model scale. In addition to the large-scale models reported in the submission, we have now evaluated **TableMaster on a smaller model, meta-llama/Meta-Llama-3.1-8B-Instruct**, to better understand the minimal model capabilities required for the framework to function effectively.
>
> Our results show that TableMaster remains highly effective even at the 7–8B scale. On our main table QA benchmark (TabFact), the plain LLM with chain-of-thought prompting reaches **74.8%** accuracy, while **TableMaster built on the same 8B backbone achieves 88.5%**, demonstrating a substantial improvement.
>
> | Method                 | Accuracy |
> |------------------------|----------|
> | LLM + Chain-of-Thought | 74.8%    |
> | TableMaster (ours)     | 88.5%    |
>
> These observations suggest that TableMaster does **not** rely on very large or proprietary models,, nor does it require sophisticated or highly advanced reasoning capabilities. Instead, it requires only **moderate LLM capabilities**, including:
> - basic instruction-following ability,
> - schema-aware interpretation of column headers,
> - stable execution of structured prompts (for column selection and SQL-style row filtering),
> - and the ability to perform light multi-step reasoning.
>
> Our results show that TableMaster delivers robust and stable performance gains on 7–8B models, with no signs of degradation compared to larger-scale LLMs.
>
> We will include this discussion and the new 8B results in the revision to clarify the minimal model requirements and demonstrate that TableMaster generalizes well across model sizes.
>
> ---

---

> ### Author Response · Authors · 2025-11-26
>
> ## **[W1 & Q2] Novelty Clarification and Differentiation from Prior Work**
>
>
> Below we clarify the novelty of TableMaster and its position relative to prior work through three components: **Detailed Comparison**, **Motivation and Contribution**, and **Overview of Novelty & Meaningful Departure from Prior Work**.
>
> ---
>
> ### **1. Detailed Comparison**
>
> We compare TableMaster with prior studies that adapt LLMs for table understanding **without fine-tuning**, which is most relevant to our setup. Our framework is motivated by four core challenges identified in Section 3, while existing works typically address only one of them.
>
> ---
>
> #### **1.1 Difficulty in Locating Target Data**
>
> These methods primarily focus on sub-table construction, decomposition, or context reduction:
> - DATER [4]
> - TabSQLify [5]
> - ReAcTable [6]
> - TAP4LLM [7]
> - Tree-of-Table [8]
> - Chain-of-Table [9]
>
> **Our findings:**
> Despite their sophisticated designs, we observe that complex sub-table construction methods often perform poorly in practice and result in high computational cost—especially Tree-of-Table and Chain-of-Table. In contrast, our experiments show that the most reliable approach is **LLM-based column selection combined with SQL-based row filtering**, which achieves better performance and efficiency.
>
> In addition, prior works rarely address the **missing-information issue** in sub-table extraction. TableMaster explicitly tackles this through a **Fallback Strategy** (Appendix J, M), where we reconstruct incomplete subtables to avoid information loss—an important issue overlooked by earlier approaches.
>
> ---
>
> #### **1.2 Table Semantic Deficiency**
>
> Few works directly tackle semantic enrichment of tables.
> Some early ideas (table-to-text verbalization) suggest potential benefits, but they are not systematically integrated into reasoning pipelines.
>
> In TableMaster, we show that **structured verbalization enriches table semantics** and consistently improves reasoning quality, addressing semantic sparsity that existing methods leave unresolved.
>
> ---
>
> #### **1.3 Numerical Inaccuracy in Textual Reasoning**
>
> These methods introduce Python/SQL execution to mitigate numerical mistakes:
> - BINDER [1]
> - LEVER [2]
> - PoTable [3]
> - MIX-SC [10]
> - SpreadsheetEncoder [11]
>
> While symbolic execution is useful, prior works implicitly assume it is always preferable.
> Our analysis reveals that **textual reasoning is more effective for most table queries**, and symbolic execution should be used selectively, primarily in calculation-intensive scenarios. TableMaster introduces **Adaptive Reasoning**, which dynamically selects between textual and symbolic reasoning pathways for stronger and more reliable performance.
>
> ---
>
> #### **1.4 Semantic Inflexibility in Symbolic Reasoning**
>
> - MIX-SC [10]
>
> MIX-SC alleviates symbolic rigidity through 10-way self-consistency (5 textual + 5 symbolic), but at the cost of substantial latency. It also does not analyze why textual reasoning helps.
>
> TableMaster provides a **single-shot, efficient** alternative via Adaptive Reasoning, matching the benefits of self-consistency without its computational overhead, and explaining *why* textual reasoning works better in many cases.
>
> ---
> **References:**
>
> [1] Z. Cheng et al., “BINDER: Binding Language Models in Symbolic Reasoning Framework,” 2023.
> [2] A. Ni et al., “LEVER: Learning to Verify Language-to-Code Generation with Execution,” ICML, 2023.
> [3] Q. Mao et al., “PoTable: Programming Standardly on Table-based Reasoning Like a Human Analyst,” 2024.
> [4] Y. Ye et al., “DATER: Large Language Models are Versatile Decomposers for Table-based Reasoning,” 2023.
> [5] M. H. Nahid and D. Rafiei, “TabSQLify: Enhancing Reasoning Capabilities of LLMs Through Table Decomposition,” NAACL, 2024.
> [6] Z. Zhang et al., “ReAcTable: ReAct-style Table Reasoning via Intermediate Tables and Code Execution,” 2023.
> [7] J. Sui et al., “TAP4LLM: Table Prompting Toolbox for Large Language Models,” 2024.
> [8] Z. Ji et al., “Tree-of-Table: Hierarchical Sub-Table Construction for Table Question Answering,” 2024.
> [9] Z. Wang et al., “Chain-of-Table: Evolving Tables in the Reasoning Chain for Table Understanding,” 2024.
> [10] Q. Liu et al., “Rethinking Tabular Data Understanding with Large Language Models,” 2024.
> [11] W. Dong et al., “SpreadsheetEncoder: Interpreting Tabular Data within Spreadsheet Environments,” 2024.
>
>
> ---

---

> ### Author Response · Authors · 2025-11-26
>
> ## **[W1 & Q2] Novelty Clarification and Differentiation from Prior Work (Cont.)**
>
> ### **2. Motivation and Contribution**
>
> LLMs struggle with tables due to structural complexity, semantic sparsity, and mixed reasoning demands. Existing works typically propose isolated improvements—better prompts, specialized retrievers, symbolic reasoning modules—but lack a **generalizable and unified framework**.
>
> TableMaster provides a **systematic, extensible recipe** that organizes:
> - table representation and semantic enrichment,
> - selective sub-table construction,
> - dynamic reasoning mode selection,
> - program execution,
> - fallback and reconstruction mechanisms,
>
> into a **cohesive and principled framework**.
> To our knowledge, TableMaster is the **first challenge-driven, holistic study** that evaluates, analyzes, and organizes LLM-based table reasoning into a unified design space.
>
> ---
>
> ### **3. Overview of Novelty and a Meaningful Departure from Prior Work**
>
> Although TableMaster uses components that have appeared independently in earlier works, it departs from them in several important ways:
>
> ---
>
> #### **3.1 A Systematic and Unified Recipe**
>
> Prior works focus on isolated improvements (e.g., a better retrieval scheme, a single reasoning pipeline, or a symbolic executor). TableMaster provides a **unified and extensible recipe** grounded in four core challenges, assembling fragmented insights into a coherent framework with broader generalization and transferability.
>
> ---
>
> #### **3.2 Mechanisms Beyond Simple Integration**
>
> We introduce new mechanisms that address practical gaps in earlier methods:
>
> - **Adaptive Reasoning (Appendix L):**
>   Dynamic switching between textual and symbolic reasoning without expensive self-consistency.
>   Prior works assume static reasoning or rely heavily on ensembles.
>
> - **Fallback Strategy in Table-of-Focus (Appendix J, M):**
>   A novel reconstruction mechanism that corrects incomplete subtables—an issue rarely handled explicitly in retrieval-based systems.
>
> ---
>
> #### **3.3 Providing Fundamental Insights Missing from Previous Work**
>
> Beyond algorithmic integration, TableMaster offers **deeper, empirical insights** that prior studies have not surfaced, including:
> - Why textual reasoning often outperforms symbolic reasoning and how to combine them,
> - Why complex sub-table extraction techniques fail in practice,
> - How semantic enrichment directly influences table reasoning quality,
> - How missing-information risks propagate through table reasoning pipelines,
> - What is the most efficient and effective way to make LLMs understand table better in each stage.
>
> These insights reshape how table reasoning should be framed for LLMs and provide conceptual guidance that earlier work lacks.
>
> ---
>
> #### **3.4 A Practical and Scalable Foundation**
>
> TableMaster is not an incremental extension; it represents a meaningful shift toward robust, efficient, and general-purpose table reasoning, offering a practical foundation for future research. Moreover, our conclusions are supported by extensive experiments across diverse datasets, reasoning types, and LLM backbones. Conducting these evaluations required substantial computational effort, further underscoring the depth, reliability, and scalability of our findings.
>
> ---
>
> Thank you again for your valuable feedback. We believe these updates strengthen the paper and adequately address the reviewer’s concerns.
>
> **Sincerely,**
> Authors of *TableMaster*

---

### Official Review · Reviewer_MV4i · 2025-11-01

**Soundness:** 3
**Presentation:** 3
**Contribution:** 2
**Rating:** 6
**Confidence:** 3

**Summary:**

This paper introduces TableMaster, a novel framework designed to enhance how Large Language Models (LLMs) understand tabular data. The research addresses four key obstacles: difficulty in data localization, semantic deficiency, numerical inaccuracies, and inflexible symbolic reasoning. TableMaster employs a multi-faceted strategy, beginning by isolating relevant data into a "table-of-focus" and then using "verbalization" to enrich it with semantic context. The framework integrates program-aided reasoning and features an adaptive mechanism that dynamically balances textual and symbolic approaches based on the query. This method has achieved state-of-the-art performance on the WikiTQ and TabFact benchmarks, notably reaching 78.13% accuracy on WikiTQ with GPT-4o-mini, significantly surpassing existing baselines.

**Strengths:**

1. The paper demonstrates strong empirical rigor through comprehensive experiments across multiple benchmark datasets and baselines. The thorough ablation studies effectively validate the contributions of individual components, providing clear evidence of the method's effectiveness.
2. The authors developed a robust system for analyzing and extracting information from general tabular data, with carefully designed modules compatible with various language model backbones.

**Weaknesses:**

1. Several key experiments are missing from the main paper, such as the analysis of adaptive reasoning. Including these results in the main body would strengthen the paper.
2. The framework is thoughtfully designed and comprehensive, but many of its sub-tasks have been extensively studied, with closely related methods already proposed. As a result, the incremental novelty appears limited.
3. In the related work section the connections of similar methods to this work are not clear. It would help to position the framework relative to each major line of work (what is shared, what differs, and why those differences matter), and to articulate the specific gaps in prior methods that this paper addresses.

**Questions:**

Please refer to the weaknesses part.

---

> ### Author Response · Authors · 2025-11-26
>
> Thank you for the reviewer’s constructive and insightful comments. We sincerely appreciate the feedback, which has helped us significantly strengthen the clarity and completeness of the paper. Below we address each concern in detail.
>
> ---
>
> ## **[W1] Position of key experiments (adaptive reasoning analysis)**
>
> Thank you for highlighting this point. The adaptive reasoning (AR) analysis is indeed an important component of our method. Due to space limitations, we placed the full ablation in the appendix, but we agree that presenting it in the main paper would strengthen the contribution. In the revision, since the camera-ready version allows additional space, we will move the AR results into the main paper and highlight the key insights from this analysis.
>
> ---

---

> ### Author Response · Authors · 2025-11-26
>
> ## **[W2 & W3] Novelty Clarification and Differentiation from Prior Work**
>
>
> Below we clarify the novelty of TableMaster and its position relative to prior work through three components: **Detailed Comparison**, **Motivation and Contribution**, and **Overview of Novelty & Meaningful Departure from Prior Work**.
>
> ---
>
> ### **1. Detailed Comparison**
>
> We compare TableMaster with prior studies that adapt LLMs for table understanding **without fine-tuning**, which is most relevant to our setup. Our framework is motivated by four core challenges identified in Section 3, while existing works typically address only one of them.
>
> ---
>
> #### **1.1 Difficulty in Locating Target Data**
>
> These methods primarily focus on sub-table construction, decomposition, or context reduction:
> - DATER [4]
> - TabSQLify [5]
> - ReAcTable [6]
> - TAP4LLM [7]
> - Tree-of-Table [8]
> - Chain-of-Table [9]
>
> **Our findings:**
> Despite their sophisticated designs, we observe that complex sub-table construction methods often perform poorly in practice and result in high computational cost—especially Tree-of-Table and Chain-of-Table. In contrast, our experiments show that the most reliable approach is **LLM-based column selection combined with SQL-based row filtering**, which achieves better performance and efficiency.
>
> In addition, prior works rarely address the **missing-information issue** in sub-table extraction. TableMaster explicitly tackles this through a **Fallback Strategy** (Appendix J, M), where we reconstruct incomplete subtables to avoid information loss—an important issue overlooked by earlier approaches.
>
> ---
>
> #### **1.2 Table Semantic Deficiency**
>
> Few works directly tackle semantic enrichment of tables.
> Some early ideas (table-to-text verbalization) suggest potential benefits, but they are not systematically integrated into reasoning pipelines.
>
> In TableMaster, we show that **structured verbalization enriches table semantics** and consistently improves reasoning quality, addressing semantic sparsity that existing methods leave unresolved.
>
> ---
>
> #### **1.3 Numerical Inaccuracy in Textual Reasoning**
>
> These methods introduce Python/SQL execution to mitigate numerical mistakes:
> - BINDER [1]
> - LEVER [2]
> - PoTable [3]
> - MIX-SC [10]
> - SpreadsheetEncoder [11]
>
> While symbolic execution is useful, prior works implicitly assume it is always preferable.
> Our analysis reveals that **textual reasoning is more effective for most table queries**, and symbolic execution should be used selectively, primarily in calculation-intensive scenarios. TableMaster introduces **Adaptive Reasoning**, which dynamically selects between textual and symbolic reasoning pathways for stronger and more reliable performance.
>
> ---
>
> #### **1.4 Semantic Inflexibility in Symbolic Reasoning**
>
> - MIX-SC [10]
>
> MIX-SC alleviates symbolic rigidity through 10-way self-consistency (5 textual + 5 symbolic), but at the cost of substantial latency. It also does not analyze why textual reasoning helps.
>
> TableMaster provides a **single-shot, efficient** alternative via Adaptive Reasoning, matching the benefits of self-consistency without its computational overhead, and explaining *why* textual reasoning works better in many cases.
>
> ---
> **References:**
>
> [1] Z. Cheng et al., “BINDER: Binding Language Models in Symbolic Reasoning Framework,” 2023.
> [2] A. Ni et al., “LEVER: Learning to Verify Language-to-Code Generation with Execution,” ICML, 2023.
> [3] Q. Mao et al., “PoTable: Programming Standardly on Table-based Reasoning Like a Human Analyst,” 2024.
> [4] Y. Ye et al., “DATER: Large Language Models are Versatile Decomposers for Table-based Reasoning,” 2023.
> [5] M. H. Nahid and D. Rafiei, “TabSQLify: Enhancing Reasoning Capabilities of LLMs Through Table Decomposition,” NAACL, 2024.
> [6] Z. Zhang et al., “ReAcTable: ReAct-style Table Reasoning via Intermediate Tables and Code Execution,” 2023.
> [7] J. Sui et al., “TAP4LLM: Table Prompting Toolbox for Large Language Models,” 2024.
> [8] Z. Ji et al., “Tree-of-Table: Hierarchical Sub-Table Construction for Table Question Answering,” 2024.
> [9] Z. Wang et al., “Chain-of-Table: Evolving Tables in the Reasoning Chain for Table Understanding,” 2024.
> [10] Q. Liu et al., “Rethinking Tabular Data Understanding with Large Language Models,” 2024.
> [11] W. Dong et al., “SpreadsheetEncoder: Interpreting Tabular Data within Spreadsheet Environments,” 2024.
>
>
> ---

---

> ### Author Response · Authors · 2025-11-26
>
> ## **[W2 & W3] Novelty Clarification and Differentiation from Prior Work (Cont.)**
>
> ### **2. Motivation and Contribution**
>
> LLMs struggle with tables due to structural complexity, semantic sparsity, and mixed reasoning demands. Existing works typically propose isolated improvements—better prompts, specialized retrievers, symbolic reasoning modules—but lack a **generalizable and unified framework**.
>
> TableMaster provides a **systematic, extensible recipe** that organizes:
> - table representation and semantic enrichment,
> - selective sub-table construction,
> - dynamic reasoning mode selection,
> - program execution,
> - fallback and reconstruction mechanisms,
>
> into a **cohesive and principled framework**.
> To our knowledge, TableMaster is the **first challenge-driven, holistic study** that evaluates, analyzes, and organizes LLM-based table reasoning into a unified design space.
>
> ---
>
> ### **3. Overview of Novelty and a Meaningful Departure from Prior Work**
>
> Although TableMaster uses components that have appeared independently in earlier works, it departs from them in several important ways:
>
> ---
>
> #### **3.1 A Systematic and Unified Recipe**
>
> Prior works focus on isolated improvements (e.g., a better retrieval scheme, a single reasoning pipeline, or a symbolic executor). TableMaster provides a **unified and extensible recipe** grounded in four core challenges, assembling fragmented insights into a coherent framework with broader generalization and transferability.
>
> ---
>
> #### **3.2 Mechanisms Beyond Simple Integration**
>
> We introduce new mechanisms that address practical gaps in earlier methods:
>
> - **Adaptive Reasoning (Appendix L):**
>   Dynamic switching between textual and symbolic reasoning without expensive self-consistency.
>   Prior works assume static reasoning or rely heavily on ensembles.
>
> - **Fallback Strategy in Table-of-Focus (Appendix J, M):**
>   A novel reconstruction mechanism that corrects incomplete subtables—an issue rarely handled explicitly in retrieval-based systems.
>
> ---
>
> #### **3.3 Providing Fundamental Insights Missing from Previous Work**
>
> Beyond algorithmic integration, TableMaster offers **deeper, empirical insights** that prior studies have not surfaced, including:
> - Why textual reasoning often outperforms symbolic reasoning and how to combine them,
> - Why complex sub-table extraction techniques fail in practice,
> - How semantic enrichment directly influences table reasoning quality,
> - How missing-information risks propagate through table reasoning pipelines,
> - What is the most efficient and effective way to make LLMs understand table better in each stage.
>
> These insights reshape how table reasoning should be framed for LLMs and provide conceptual guidance that earlier work lacks.
>
> ---
>
> #### **3.4 A Practical and Scalable Foundation**
>
> TableMaster is not an incremental extension; it represents a meaningful shift toward robust, efficient, and general-purpose table reasoning, offering a practical foundation for future research. Moreover, our conclusions are supported by extensive experiments across diverse datasets, reasoning types, and LLM backbones. Conducting these evaluations required substantial computational effort, further underscoring the depth, reliability, and scalability of our findings.
>
> ---
>
> Thank you again for your valuable feedback. We believe these updates strengthen the paper and adequately address the reviewer’s concerns.
>
> **Sincerely,**
> Authors of *TableMaster*

---

### Meta-Review · Area_Chair_jhU6 · 2025-12-29

**Summary:**

This paper presents TableMaster, a framework for improving table understanding with LLMs. The authors clearly identify four recurring challenges that include data localization, weak table semantics, numerical errors, and rigid symbolic reasoning. These are addressed through a cohesive pipeline combining prior work.  Reviewers consistently note the strength of the empirical evaluation, which covers multiple benchmarks and model sizes, including both proprietary and open-source LLMs.

Although the individual components build on prior work, the main contribution lies in how they are integrated and analyzed.  The rebuttal further clarifies this and adds missing experiments that strengthen the pap

**Reviewer Concerns:**

MV4i moderately addressed especilly about which results are in the appendix vs body of the main paper. The reviewer points out that system is stitching together several prior art and ablations are not presented to lead to a comprehensive review.

oMBW's comments about ablations, latency are addressed. However, novelty is still a concern.

sJMb's comments about using different sized base models are updated.

**Reviewer Scores:**

All reviewer scores would move up by +1

---

### Decision · Program_Chairs · 2026-01-26

Accept (Poster)